# Global methane emissions from rivers and streams

Gerard Rocher-Ros[1,2,3✉], Emily H. Stanley[4], Luke C. Loken[5], Nora J. Casson[6], Peter A. Raymond[7], Shaoda Liu[7,8], Giuseppe Amatulli[7] & Ryan A. Sponseller[1]

Methane ($CH_4$) is a potent greenhouse gas and its concentrations have tripled in the atmosphere since the industrial revolution. There is evidence that global warming has increased $CH_4$ emissions from freshwater ecosystems[1,2], providing positive feedback to the global climate. Yet for rivers and streams, the controls and the magnitude of $CH_4$ emissions remain highly uncertain[3,4]. Here we report a spatially explicit global estimate of $CH_4$ emissions from running waters, accounting for 27.9 (16.7–39.7) Tg $CH_4$ per year and roughly equal in magnitude to those of other freshwater systems[5,6]. Riverine $CH_4$ emissions are not strongly temperature dependent, with low average activation energy ($E_M = 0.14$ eV) compared with that of lakes and wetlands ($E_M = 0.96$ eV)[1]. By contrast, global patterns of emissions are characterized by large fluxes in high- and low-latitude settings as well as in human-dominated environments. These patterns are explained by edaphic and climate features that are linked to anoxia in and near fluvial habitats, including a high supply of organic matter and water saturation in hydrologically connected soils. Our results highlight the importance of land–water connections in regulating $CH_4$ supply to running waters, which is vulnerable not only to direct human modifications but also to several climate change responses on land.

Freshwater ecosystems are responsible for nearly half of global $CH_4$ emissions to the atmosphere[4,7]. Yet, among freshwaters, the role of rivers and streams in the global $CH_4$ cycle remains unclear although current best estimates of global fluvial emissions[3,4] are similar in magnitude to other important $CH_4$ sources such as biomass burning and rice cultivation[8]. Fluvial ecosystems play key parts in connecting terrestrial, marine and atmospheric carbon pools[8], and are unique in their potential to produce $CH_4$ internally, while also receiving and emitting large amounts of $CH_4$ generated externally in adjacent soils and wetlands[9,10]. Thus, global $CH_4$ emissions from streams and rivers may be regulated by multiple environmental factors that operate across land–water boundaries. Resolving these controls should improve our predictions of riverine $CH_4$ emissions and our broader understanding of how running waters process and deliver carbon to downstream ecosystems in response to climate warming and other global environmental changes.

Despite their potential as an important atmospheric source, current syntheses of riverine $CH_4$ emissions highlight extreme spatial and temporal variability, with measured rates spanning seven orders of magnitude[3,4], as well as strong fine-scale controls over $CH_4$ dynamics[10,11]. Thus, efforts to generate global estimates have been based on a simple averaging of measured $CH_4$ emissions, which has resulted in massive uncertainty[3,4,7,12], unknown global patterns[3] and large discrepancies between bottom-up inventories and top-down estimates[4,7]. Further complications arise from the fact that aquatic $CH_4$ emissions occur by diffusion and by the even-more variable process of ebullition, in which $CH_4$-rich bubbles are released from sediments. To address these uncertainties and advance our understanding of $CH_4$ dynamics in running waters, we leveraged a $CH_4$ database[13] (Global River Methane database (GRiMeDB)) containing more than 24,000 observations of $CH_4$ concentration and more than 8,000 observations of $CH_4$ fluxes (Extended Data Fig. 1) to model $CH_4$ concentrations globally using random forest machine-learning models. From these models, we can explain a substantial fraction of the total variability in $CH_4$ concentrations ($R^2$ from log-transformed modelled versus withheld observations of 0.45–0.68; Extended Data Fig. 2) and produce a seasonally and spatially explicit global estimate of $CH_4$ emissions from rivers and streams. More importantly, using this database and model outputs, we are able to identify the main drivers of $CH_4$ concentrations and fluxes from running waters across the globe.

## Global $CH_4$ patterns in rivers

Global patterns of $CH_4$ concentration in rivers and streams (Fig. 1a and Extended Data Fig. 3) highlight the influence of multiple factors that regulate the in situ production and/or supply from surrounding catchments. The highest concentrations occur in tropical biomes, which reflect elevated $CH_4$ reported in Southeast Asia[14], the Congo Basin[15] and the floodplains of the Pantanal and Amazon rivers[16]. However,

[1]Department of Ecology and Environmental Science, Umeå University, Umeå, Sweden. [2]Department of Forest Ecology and Management, Swedish University of Agricultural Sciences, Umeå, Sweden. [3]Integrative Freshwater Ecology Group, Centre for Advanced Studies of Blanes (CEAB-CSIC), Blanes, Spain. [4]Center for Limnology, University of Wisconsin–Madison, Madison, WI, USA. [5]Upper Midwest Water Science Center, United States Geological Survey, Madison, WI, USA. [6]Department of Geography, University of Winnipeg, Winnipeg, Manitoba, Canada. [7]School of the Environment, Yale University, New Haven, CT, USA. [8]State Key Laboratory of Water Environment Simulation, School of Environment, Beijing Normal University, Beijing, China. ✉e-mail: gerard.rocher.ros@slu.se

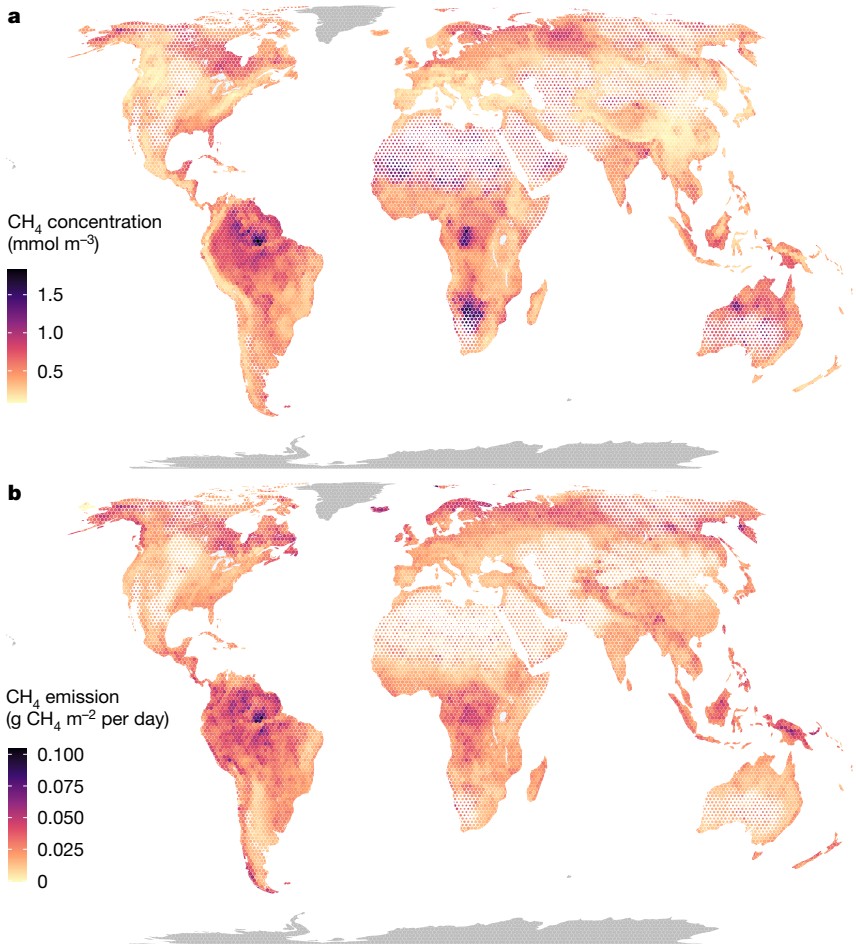

**Fig. 1 | Global patterns of CH$_4$ in rivers and streams. a,b,** Modelled yearly average CH$_4$ concentrations (**a**) and emissions (**b**) in rivers and streams. Data have been aggregated in hexagonal bins, and the size of each hexagon is rescaled with runoff, to better visualize patterns in areas with high coverage of running waters. Areas with runoff greater than 1,500 mm per year have full-sized hexagons; hexagons in areas with runoff of 500 mm per year have been reduced by 10%; and hexagons with a runoff less than 50 mm per year have been reduced by 50%. The model could not be applied in Greenland and Antarctica, which are shown in dark grey.

concentrations are also relatively high in Arctic and boreal biomes, with some of the highest global values reported from Fennoscandia, Alaska in the USA and Eastern Siberia[13]. This seemingly counterintuitive pattern—with elevated concentrations in both warm and cold regions—is consistent with our understanding of CH$_4$ production and supply in running waters[3], which is maximized when large stores of soil organic matter intersect with water-saturated environments to create anoxic conditions. In the tropics, these conditions are supported by high rates of terrestrial above-ground primary production, soil respiration and precipitation, combined with strong connectivity to adjacent wetlands[9]. In our model, all of these variables emerge as important predictors for CH$_4$ concentration from running waters (Fig. 2a). At high latitudes, CH$_4$ production is instead fuelled by large soil organic carbon stocks, extensive peatland cover and shallow groundwater tables—all variables that were also important in the model (Fig. 2a). Taken together, these results show how multiple combinations of climate and edaphic features can create the conditions for soil or sediment methanogenesis that shape global patterns of riverine CH$_4$ concentrations and emissions.

Despite clear patterns in CH$_4$ concentrations at the global scale, the most important variables in our random forest models reflect the physical template of the local landscape (Fig. 2a). Physical catchment variables such as river slope, elevation and gas-transfer velocity all have negative effects on modelled CH$_4$ concentrations. These effects are expected given that a higher slope and gas-transfer velocity favour

gas exchange between the water and the atmosphere[17], preventing the build-up of aquatic carbon gasses[18,19]. Elevation may similarly capture the turbulent nature of mountain streams but also probably signifies reduced catchment productivity or organic matter stocks in high-elevation areas[18]. The influence of these geomorphological variables in shaping CH$_4$ concentrations in rivers further highlights connections between fluvial ecosystems and anoxic environments with a high potential to generate CH$_4$ (ref. 3), as well as the turbulent nature of running waters that promotes emissions to the atmosphere. Collectively, our results suggest a set of climatic and biological variables that regulate the production and availability of CH$_4$ in running waters at global scales, with a second set of geomorphological and physical variables that regulate concentrations at river-reach scales. Importantly, individual studies highlight an even finer scale of spatial and temporal variability than is considered here[10,11,20], arising from patchiness in groundwater inputs[10] and in sediment properties[11], from fluctuations in river discharge[21], and even diel variability in factors that regulate the balance of CH$_4$ production, oxidation and flux[22]. These local controls are not captured by our model, which is based on relatively coarse spatial predictors applied to monthly aggregated CH$_4$ concentrations. Such controls probably drive the substantial fraction of unexplained variability in our model (Extended Data Figs. 2 and 4), indicating an unresolved discrepancy between global models and reach-scale field studies.

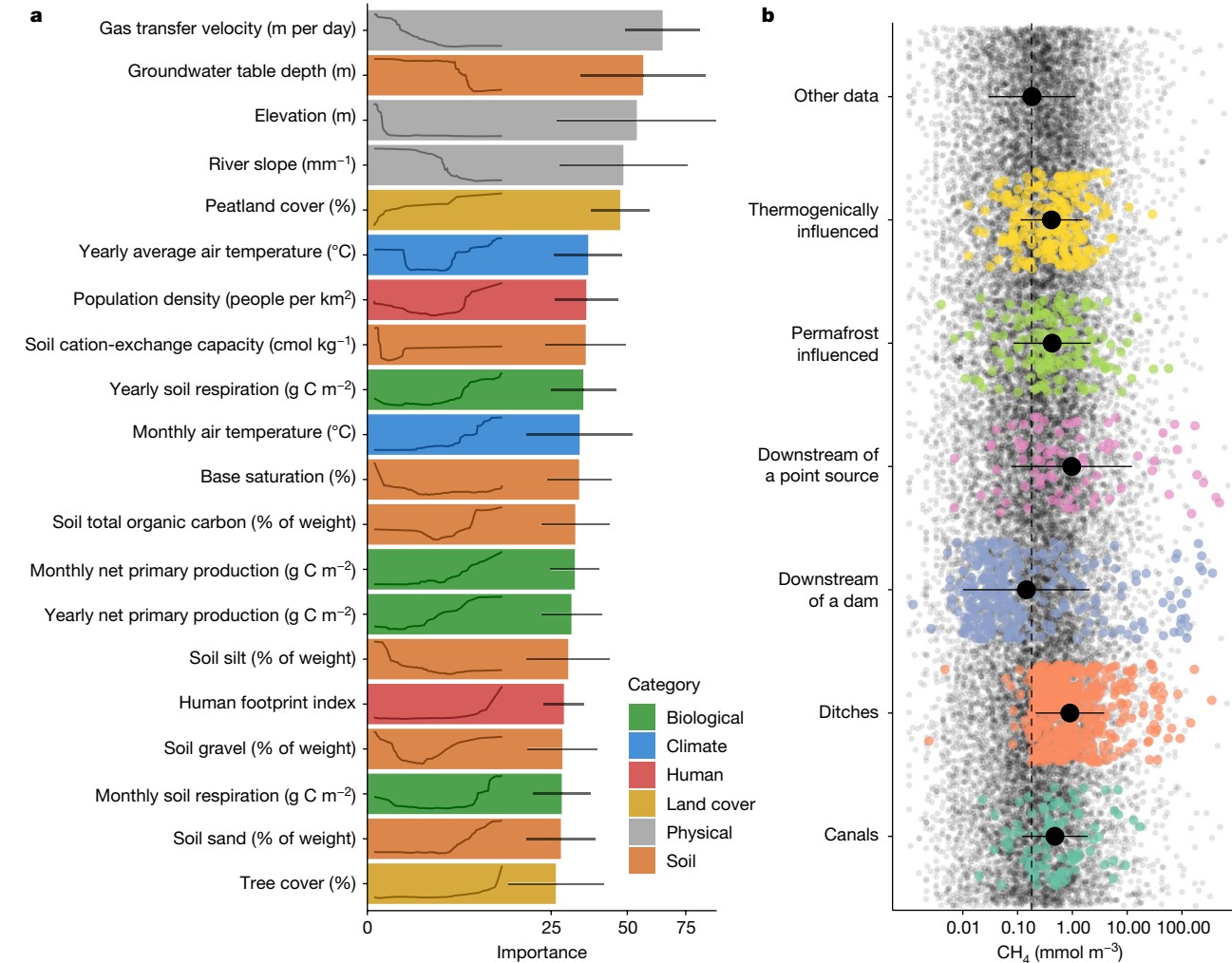

**Fig. 2 | Main drivers of CH₄ concentrations in streams. a**, The 20 most important variables in the random forest model. The $x$ axis shows the median importance across all monthly models ($n = 12$), with error lines representing standard deviation (s.d.); note the square-root transformation of the $x$ axis. The line inside each bar is the partial dependence, which represents the marginal effect of a given feature ($x$ axis) on predicted CH₄ concentrations ($y$ axis). These lines are a simplification of a more detailed version (Supplementary Information).

**b**, CH₄ concentrations of some site categories from GRiMeDB[13] were excluded from the model as they were not captured in the hydrological model or were targeted observations not representative of catchment properties (Methods). The underlying jittered points represent all other observations in GRiMeDB, with the dashed line representing the average. Each category is colour-coded, with the black dot and a line representing the mean ± s.d.

Global patterns of diffusive CH₄ emissions were similar to the patterns of concentrations (Fig. 1b), but upscaled fluxes are sensitive to the water–air gas-transfer-rate corrections performed a posteriori, particularly in mountain areas (Methods and Extended Data Fig. 5). Regardless, owing to high CH₄ concentrations and extensive riverine area, the tropics (10° S–10° N) account for the largest share of global emissions (37%), with temporal patterns that reflect shifts between wet and dry seasons (Fig. 3). However, Arctic and northern boreal areas (latitude > 50° N), despite being ice-covered during a large part of the year, contribute almost equally (17%) to the annual CH₄ emissions as temperate and subtropical latitudinal bands (15%) (30–50° N) (Fig. 3). There is also marked seasonality at high latitudes because of differences in open water-surface area and hydrological connectivity between winter and summer (Extended Data Fig. 3), and because our estimate assumes that ice or snow cover prevents riverine CH₄ emissions (Extended Data Fig. 1). This assumption is probably conservative, as CH₄ concentrations build up under ice, leading to high rates of evasion in places where channels are open or during ice break-up[23]. Importantly, in these high-latitude landscapes, rapid climate change has the potential to further increase riverine CH₄ emissions, given

ongoing decreases in river ice cover that is lengthening the open water season[24] and a projected increase in precipitation[25] that could enhance the flooded fraction of landscapes and flush CH₄ and other carbon compounds downstream. Furthermore, the thawing of frozen soils can result in high CH₄ losses to streams[26], which we detect in elevated stream CH₄ concentrations observed below the thaw slumps (Fig. 2b). Although northern biomes may be particularly vulnerable to such climate changes, shifts towards drier or wetter conditions are likely to alter the landscape-scale production and supply of CH₄ in riverine systems in any regional setting. In any case, the latitudinal patterns in emissions shown here highlight not only tropical streams and rivers as important emitters of CH₄ to the atmosphere but also the potential for northern ecosystems to have increasing contributions as a result of global climate change.

## Role of temperature and humans

Elevated CH₄ concentrations and emissions in both warm and cold regions are in apparent disagreement with the universal temperature dependence of CH₄ emissions observed among freshwater systems[1].

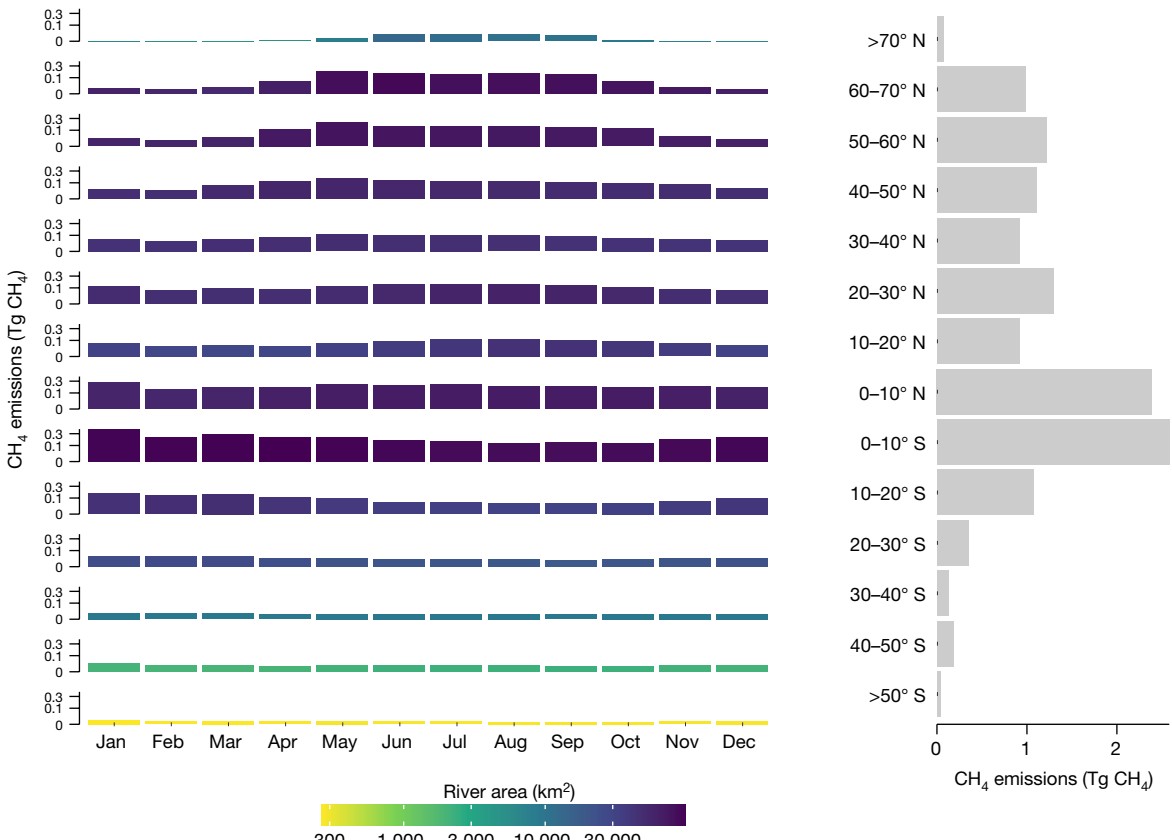

**Fig. 3 | Seasonal patterns of CH$_4$ emissions.** Left: total monthly CH$_4$ emissions for each latitudinal band (10° bins), with the colour representing total river area. Right: total yearly emissions for each latitudinal band. In the left panel, the $y$ axis is square-root transformed, and the colour scale is log transformed.

Aquatic methanogenesis is consistently and strongly regulated by temperature, with high activation energies ($E_M$) observed in culture ($E_M = 1.1$ eV) and sediment incubations ($E_M = 0.93$ eV)[1], including from fluvial environments ($E_M = 1.1$ eV)[27]. Across aquatic ecosystems and wetlands, this temperature dependence can translate into quantitatively similar thermal scaling for emissions to the atmosphere ($E_M = 0.96$ eV)[1]. Yet, the apparent temperature sensitivity of diffusive emissions from global rivers is markedly lower than these values when considered across the whole dataset ($E_M = 0.17$ eV), in individual sites (Fig. 4a) and for ebullition losses (Extended Data Fig. 6). Synthesis of our site-specific $E_M$ for rivers shows significantly lower values (median $E_M = 0.14$ eV; interquartile range = −0.16 to 0.51) compared with lakes, wetlands and rice paddies[1] ($P < 0.001$, Wilcoxon rank-sum test; Fig. 4b). We attribute these low $E_M$ estimates to the fundamentally open nature of running waters, in which external inputs not only account for a large fraction of carbon gases evaded to the atmosphere[8] but also fuel aquatic metabolic processes through terrestrial organic matter supply[28]. These strong external sources and controls dampen strict thermal sensitivity of emissions in running waters[29], particularly when compared with lakes and wetlands where metabolic processes are often more internally regulated. Furthermore, the lack of a strong thermal influence could reflect parallel increases in CH$_4$ oxidation as temperatures increases[30], as well as the potential for methanogens in groundwater to adapt to thermal stability and thus be less responsive to temperature changes[31]. In any case, although temperature is among the important predictors in our model and can be important in individual rivers[20,21], it does not operate as a first-order control over global patterns of emissions. In fact, despite much focus on the temperature dependence of aquatic methanogenesis[1,2], we suggest that the most important effects of climate change for riverine CH$_4$ emissions will probably occur through the indirect influences of warming and precipitation change on the capacity of soils and wetlands to generate CH$_4$, on the strength of hydrological connections between these sources and river channels, and on direct loading of organic matter and nutrients that can enhance near-channel and/or internal CH$_4$ production.

Besides climatic, biological and physical drivers, human population density also positively influenced CH$_4$ concentrations in our model. Humans affect multiple facets of fluvial ecosystems, many of which have the potential to enhance CH$_4$ production and/or emissions. For instance, impoundments can produce and export large masses of CH$_4$ downstream[32]; agricultural areas are sources of fine sediments, organic carbon and nutrients that promote internal CH$_4$ production[27]; and polluted waters in urbanized areas are often hotspots of CH$_4$ production[14,33]. Critically, we excluded observations from the most highly modified systems from our models because these sites are not represented by the spatial predictors used (Methods). However, CH$_4$ concentrations were often elevated in streams directly affected by wastewater treatment plants (point sources), in forest and agricultural ditches, in urban canals and in rivers affected by natural gas extraction (Fig. 2b). It is also important to note that CH$_4$ emissions from reservoirs and impoundments are not included in this study because they are usually classified as lentic waters. Nonetheless, these habitats represent human alterations of river networks and account for a large share (about 10%) of freshwater emissions[4]. Overall, these results indicate an increasing role of human activities in enhancing riverine CH$_4$ emissions and understanding and reducing such losses represents an opportunity to mitigate climate change.

## Global magnitude of riverine CH$_4$ emissions

Our estimate of annual diffusive CH$_4$ emissions to the atmosphere accounts for 13.4 (10.1–16.8) Tg CH$_4$ per year (parenthetical values

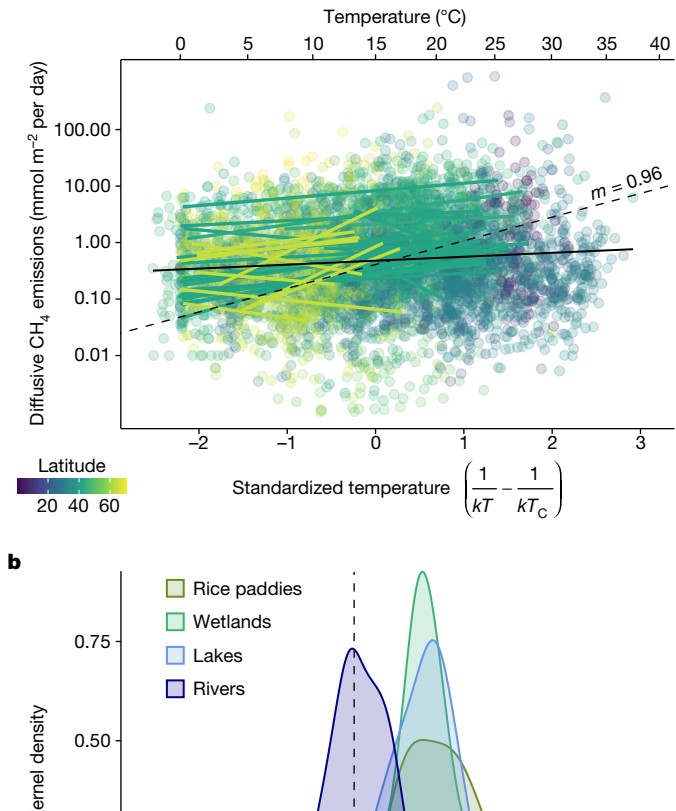

**a**

**b**

**Fig. 4 | Temperature dependence of diffusive CH₄ emissions. a**, The relationship of water temperature with measured diffusive $CH_4$ emissions in GRiMeDB[13] coloured by latitude. The solid coloured lines are linear fits for sites that have more than 20 observations and the black solid line is the linear model for all data. The dashed black line represents the slope from the average activation energy of other aquatic systems from ref. 1. The $x$ axis shows the standardized temperature following ref. 1, where $k$ is the Boltzmann constant, $T$ is the water temperature in kelvin and $T_c$ is 15 °C. The colour scale shows the absolute values of latitudinal decimal degrees. **b**, A kernel density plot ($y$ axis represents the relative number of observations) of the apparent activation energy—that is, the slope of the lines in **a**—for rivers (this study) and other freshwater systems compiled in ref. 1. The vertical dashed line shows the zero $x$ intercept.

represent the 10th–90th percentiles from Monte Carlo simulations). However, in other freshwater systems, ebullitive $CH_4$ fluxes are often the largest pathway for emissions[5,6]. The data compiled in GRiMeDB[13] highlight the paucity of ebullitive measurements from running waters and show a large range in flux rates (Extended Data Fig. 7). Literature observations similarly show extreme spatial and temporal variability in bubble-mediated $CH_4$ fluxes[20], indicating that further measurements of ebullitive emissions of $CH_4$ could support a more robust global quantification (Supplementary Information). Regardless, the dataset shows that diffusive and ebullitive fluxes in rivers are of the same magnitude, with a median of 0.157 mmol m⁻² per day and 0.128 mmol m⁻² per day, respectively, are linearly related and close to the 1:1 line (log-transformed data; Extended Data Fig. 7). This similarity suggests that diffusive and ebullitive pathways share a common source or common set of drivers. Thus, using the linear model in Extended Data

Fig. 7 as an initial estimate, we expect $CH_4$ emissions by ebullition in rivers to account for 14.5 (6.6–22.9) Tg $CH_4$ per year to the atmosphere, in total contributing to about 27.9 (16.7–39.7) Tg $CH_4$ per year. This overall annual estimate is at the higher end of previous estimates of riverine $CH_4$ emissions that ranged from 1.5 Tg $CH_4$ per year to 31 Tg $CH_4$ per year (refs. 3,4,7,12), and is similar in magnitude to lakes and reservoirs[5,6]. Our spatially and temporally explicit estimate of $CH_4$ emissions has substantially less uncertainty than past efforts (Extended Data Fig. 8) as it is based on orders of magnitude more data, making it more suitable for inclusion in the global $CH_4$ budget[7].

Our analysis of the GRiMeDB database[13] shows that $CH_4$ emissions from streams and rivers are globally important but are influenced by fundamentally different sets of drivers when compared with other freshwater ecosystems. For instance, temperature is usually a first-order control on aquatic $CH_4$ production and is used as a main parameter in process-based models predicting emissions from wetlands[34] and lakes[35] and projecting future emissions under climate change. By contrast, for running waters, the lateral inputs of $CH_4$ from wetlands and soils seem substantial, and thus the indirect effects of global change operating beyond conventional aquatic ecosystem boundaries seem to regulate emissions much more strongly than the direct, internal effects. Therefore, future process-based models that attempt to represent riverine $CH_4$ emissions may be improved by focusing on the processes behind the landscape and hydrological drivers suggested in this study. Furthermore, this dependence on external processes, together with observations from the most highly modified aquatic ecosystems, highlights the potential for humans to influence $CH_4$ emissions from running waters, enabling concrete measures for climate mitigation that could reduce emissions of such an important greenhouse gas.

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

## Methods

### Underlying global hydrology

To upscale global riverine $CH_4$ emissions, we estimated the river surface area and the gas transfer velocity across the global river network following ref. 36. In brief, we used the Global Reach-Scale A Priori Discharge Estimates (GRADES), which represents the global river drainage network with modelled daily discharge since 1979 (ref. 37). GRADES contains around 2.9 million river reaches, with a median reach length of 6.8 km (Supplementary Table 1). To capture seasonal patterns in river hydrology, we summarized monthly average discharge $Q$ and its coefficient of variation CV for each river reach. Using this monthly discharge, we estimated the velocity $V$ (m s$^{-1}$) and width $W$ (m) from established scaling relationships following ref. 36. We could then estimate river surface area for every reach and month as the product of the seasonally varying $W$ and reach length. The river surface area was further corrected for periods when rivers are either dry or covered by ice, with procedures described in ref. 36. Here we assumed that those conditions prevent rivers from emitting $CH_4$ to the atmosphere. This is a conservative assumption given that ice cover in rivers is often discontinuous and $CH_4$ may still be emitted during cold seasons[38], and dry riverbeds can emit $CH_4$, albeit typically at lower rates than when water is present[39]. To estimate the gas-transfer velocity ($k_{600}$) we used equation 5 of ref. 17—which is used in ref. 36 and in other scaling studies—as this provides reasonable estimates across ranges of slope and water velocity. The equation is

$$k_{600} = S \times V \times 2{,}841 \pm 107 + 2.02 \pm 0.209 \tag{1}$$

where $S$ (m m$^{-1}$) is the river-reach slope obtained from GRADES, and the coefficients are mean ± s.d.

Channel initiation in the GRADES network begins at a catchment size of approximately 25 km$^2$, which results in average widths for first-order streams of around 4 m, and thus misses the smallest channels in most drainage systems. For the smallest streams not captured by GRADES, we extrapolated the stream length and width to calculate the extrapolated river area. To accurately extrapolate river network properties, it is important to first group basins with similar properties in terms of climate, and we followed the same procedure as in ref. 36 using 78 basins that share a common set of hydroclimatic properties. The extrapolation was done from the smallest stream order available up to stream widths of 0.3 m, which is the median stream width of the smallest streams measured across multiple catchments[40]. For those streams not captured in GRADES, we assigned the $k_{600}$ and $CH_4$ concentration of the first-order streams of the basin to estimate $CH_4$ emissions.

### CH$_4$ concentrations

The measured riverine $CH_4$ concentrations were obtained from the GRiMeDB[13]; data are available at the Environmental Data Initiative[41]. GRiMeDB contains 24,024 observations of $CH_4$ concentrations from 5,037 sites distributed globally (Extended Data Fig. 1), as well as 8,129 direct observations of $CH_4$ diffusive flux and 620 observations of ebullitive flux. Observations from targeted sites (that is, locations affected by fracking, below dams and below wastewater treatment plants) or locations not captured in the hydrological dataset used (for example, ditches or glacial termini) were excluded from the modelling, resulting in 19,440 observations in our analysis. The reason to exclude those sites is that the measured $CH_4$ concentrations are heavily affected by those features, but those features are not captured in the catchment properties used for the model. All sites in GRiMeDB were snapped to the closest river reach in GRADES. Several reaches in GRADES had a large number of assigned GRiMeDB observations, occurring in areas with long time series or intensive spatial surveys. In those cases, the average $CH_4$ concentration of all sites was used to represent a single GRADES river reach. Because data aggregation can introduce bias in estimates, we use median values instead of means to dampen the influence of extreme values. For sites where observations spanned multiple years, data were aggregated monthly using the average value. After this temporal and spatial aggregation, we had 6,503 observations that were used for the modelling. We are aware that this data aggregation procedure can result in relationships at large spatial scales that may be different when assessing the same relationships at finer spatial scales, the so-called ecological fallacy, but this was a necessary step to quantify and understand river emissions at global scales.

### Predictor variables

We used an array of spatially explicit global datasets to predict riverine $CH_4$ concentrations. Variables include multiple aspects of the surface properties of Earth, including physical (slope, elevation), climatic (temperature, precipitation), land cover and soil properties, as well as potential human impacts (Supplementary Table 2). As we aimed to characterize the seasonal dynamics of $CH_4$ emissions, predictors with seasonal variability were used at a monthly time scale. We assigned all of those predictors to the corresponding GRADES reach. Most predictors were at a 1-km spatial resolution, and for those variables, we calculated the average value for the sub-catchment of each GRADES reach. Coarser predictors (spatial resolution of 10–55 km) were simply assigned to the underlying GRADES reach. For modelling purposes, we used the subset of GRADES that had a corresponding observation in GRiMeDB. For time-varying predictors (Supplementary Table 2), we assigned the monthly value for the month when the $CH_4$ observation was taken. Some variables that were highly skewed (Supplementary Table 2) were log transformed before modelling.

### Random forest modelling

We used random forest models to predict $CH_4$ concentrations and to understand the main drivers (but other machine-learning models such as XGBoost and a neural network were also explored; see Supplementary Information). Modelling was performed using the ranger package of R (v.0.13.1)[42] under the tidymodels environment (v.0.1.4)[43]. To capture seasonal patterns, we constructed a different random forest model for each month, but first we explored all data together to find the optimal model structure. To do this, we first removed several independent variables that were highly correlated with others (Pearson's $r > 0.95$). We retained the variable that best captured the whole ecosystem status or processes. Variables removed were as follows: gross primary production, because it was closely related to net primary production; heterotrophic and autotrophic respiration, because these were strongly related to total soil respiration; and nitrogen load from agriculture, aquaculture and point sources, because these were closely related to phosphorus inputs. Soil properties used also contained a value for the top and bottom soil layers, which were highly related, and so we retained only the top soil properties for modelling.

We then proceeded to find the best model structure by tuning the model. Random forests depend largely on three parameters: the number of variables to use in each split (mtry), the number of trees (n_trees) and the minimum size of data points before splitting a tree (min_n). This tuning was performed by first doing a grid search, in which combinations of a broad range of min_n (0–40), mtry (0–50) and n_trees (500–2,000) values were explored to find the best-performing model, implemented in a 10-fold cross-validation. After the initial broad grid search was performed, this process was repeated with a narrower range of values with which the model performed better. Once the best combination of min_n and mtry was found, a third search was performed to find the optimal n_trees parameter. The hyperparameters finally selected are mtry = 13, min_n = 8 and n_trees = 1,200.

A model for each month was then constructed with the selected hyperparameters. For each model, we used observations from a given month as well as the adjacent months (for example, for February we used data collected in January, February and March). We used this

approach to have a larger number of observations, as well as to guarantee a stronger coherence among models, given that the spatial coverage also varies across months (Extended Data Fig. 1), while still capturing any seasonal patterns. For each model, 80% of the observations were used for training and 20% were withheld for testing. The model performance was then assessed with the training data using $R^2$ and root mean squared error as performance metrics. Here, the $R^2$ ranged from 0.45 to 0.68, and the root mean squared error ranged from 1.75 mmol m$^3$ to 2.23 mmol m$^3$ (Extended Data Fig. 1). Model residuals against predictions were also assessed visually and in some months there was a weak tendency to overestimate concentrations at high values (Extended Data Fig. 2).

Model results were further explored by assessing the importance of different variables and the partial dependence of these variables on the predicted CH$_4$ concentrations. Variable importance in the random forest models was estimated using the mean decrease in accuracy using the vip package in R (v.0.3.2)[44]. This was performed for every monthly model and is summarized in Fig. 2 as the median and standard error of the mean decrease in accuracy using these 12 models. To explore the marginal effect of each variable on the modelled CH$_4$ concentrations, we created partial-dependence plots using the DALEX package in R (v.2.3.0)[45]. This was done on the basis of yearly averages for each site instead of monthly averages to obtain one plot per variable. Those plots are found in the Supplementary Information, but an abstracted version that just captures the shape of the relationship is also shown in Fig. 2 (inset bars).

We then used these random forest models to upscale CH$_4$ concentrations globally. This was done by using the monthly models to predict CH$_4$ concentrations for each GRADES river reach. We also quantified the spatially explicit uncertainty using the infinitesimal jackknife method[46] implemented in ranger[42], which produces an s.d. of the mean for each river reach and month. The maps of monthly modelled concentrations are shown in Extended Data Fig. 3 and the average map of the modelled s.d. is shown in Extended Data Fig. 4. We also assessed whether the dataset of observations is representative of the globe, or whether there are locations where the model is extrapolating outside the observed domain. This was performed following procedures described in ref. 47, with a detailed explanation in GRiMeDB[13]. In this case, the procedure was repeated for every month (Extended Data Fig. 1, red polygons).

## Upscaling diffusive CH$_4$ fluxes

CH$_4$ emissions from a given reach are the product of the river-reach area and the CH$_4$ flux rate to the atmosphere. Diffusive CH$_4$ flux rates were estimated using Fick's law as the product of the gas-transfer velocity ($k_{600}$) and the excess of CH$_4$ concentration in water following ref. 36. First, the modelled $k_{600}$, standardized for a Schmidt number of 600, was converted to $k_{CH_4}$ (for the corresponding Schmidt number of CH$_4$ and modelled water temperature) using the equation and tables in ref. 17. Water temperature was modelled from air temperature for each river reach and month following ref. 17. Excess CH$_4$ in the water was estimated as the difference in concentration of the modelled CH$_4$ concentration and the expected CH$_4$ concentration if the water was in equilibrium with the atmosphere. We used an atmospheric CH$_4$ concentration of 1.83 ppm by volume, which is the average of the period 2010–2020 in the global mean CH$_4$ atmospheric concentration (data from the National Oceanic and Atmospheric Administration[48]) when more than 74% of the samples in GRiMeDB were taken. This partial pressure of CH$_4$ was converted to mmol CH$_4$ m$^{-3}$ using the modelled water temperature as well as the corresponding atmospheric pressure derived from site elevation. Monthly CH$_4$ emissions were then calculated as the product of the CH$_4$ flux rate and the effective river area for each reach.

## Uncertainty and refinement of the estimate

We quantified the uncertainty in the CH$_4$ flux estimate using Monte Carlo simulations, including uncertainty from the three main drivers

of the flux: the $k_{600}$, the modelled CH$_4$ concentration and the estimated river surface area. For $k_{600}$, we used the s.d. from the model in ref. 17 (equation (1)). For CH$_4$, we used the s.d. of the mean estimated for each GRADES reach using the procedure described above (Extended Data Fig. 4). For river area, we used the uncertainty in river discharge for each reach, which is the CV of the daily discharge values from GRADES. Given that width was estimated from discharge, the CV from discharge was assumed to be very similar to the CV of width. The Monte Carlo procedure was performed for each river reach, using normal distributions with the parameters described above and repeated 1,000 times. We report uncertainty from the Monte Carlo model as the 5th–95th percentiles of the resulting flux distribution. We also assessed the importance of each of the three parameters by performing a one-at-a-time Monte Carlo sensitivity analysis. This was done by increasing or decreasing 1 s.d. of the mean value of a given parameter in each river reach. We repeated the experiment six times, one for each parameter, and for each parameter with an increase and a decrease of 1 s.d. Results of the sensitivity analysis are summarized in Extended Data Fig. 8.

When initially calculating fluxes, global patterns were characterized by extremely high fluxes for streams in mountainous landscapes (Extended Data Fig. 9a). However, rivers with high turbulence and thus high $k$ should have low CH$_4$ concentrations, virtually close to atmospheric equilibrium as the gas fluxes in these areas are limited by the source[18,19]. This reality is shown in the global model, which highlights the strong importance of river slope and $k$ as drivers of concentration (Fig. 2), which results in low modelled CH$_4$ concentrations in mountain regions (Fig. 1a). Despite this relationship, when calculating fluxes to the atmosphere, mountain areas had markedly higher rates than the rest of the world, as well as higher values than are reported in the literature for these regions[18,49]. These high values indicate a modelling artefact that occurs when estimating fluxes at large scales as the product of excess CH$_4$ concentrations measured locally but applied to a $k$ value obtained at a larger scale, resulting in a total flux to the atmosphere larger than the stock available in the water. This mismatch in scale can create a situation in which, even at low CH$_4$ concentrations, water-to-atmosphere fluxes seem to be driven by extremely high reaeration rates (that is, are transfer limited), which is inconsistent with our empirical understanding of carbon gas emissions from streams[18,19].

We applied three separate approaches in an effort to minimize this artefact in our estimates of global CH$_4$ emissions (Extended Data Fig. 5): (1) we capped $k$ at 35 m day$^{-1}$, a threshold at which bubble-mediated fluxes begin to dominate emissions[50]; (2) we eliminated the most extreme emission values by capping fluxes above 2 s.d. of the mean; and (3) we reduced $k$ in specific river reaches in such a way that fluxes were regulated by supply (that is, the CH$_4$ concentration) rather than reaeration. The results of the three options are shown in Extended Data Fig. 5.

To implement option 3, we first estimated the 95% gas footprint length for each river reach—that is, the distance upstream in which 95% of the CH$_4$ would evade because of advection and transport in the absence of other inputs. This footprint length $F_L$ (m) is estimated as

$$F_L = (3 \times V)/K \tag{2}$$

following ref. 51, where $K$ is the gas reaeration coefficient (per day), calculated as $K = k/D$ (where $D$ is the river depth). We interpret a result of $F_L \ll R_L$ (where $R_L$ is the reach length in m) to indicate that the modelled CH$_4$ concentration cannot be maintained for the whole reach because of the high exchange with the atmosphere, resulting in a modelled flux that is larger than the pool size available for evasion. The correction for this involved decreasing $k$ such that $F_L = R_L$, so that within a given reach, the entire pool of CH$_4$ can be evaded to the atmosphere, but not more. This correction is implemented by rearranging equation (2) as

$$k = (3 \times V \times D)/R_L \tag{3}$$

Note that $V$ and $D$ are estimated using the scaling relationships with discharge as given in ref. 17, and vary from month to month for a given reach because of the changes in discharge. This correction affected 20% of the river reaches and preferentially targets mountain reaches with high channel slopes. Furthermore, it produces a global map that more closely resembles the patterns in concentrations and is more consistent with our mechanistic understanding of gas evasion and principles of mass balance. We compared the relationship between $CH_4$ concentrations and fluxes from these three options with the empirical flux observations in the GRiMeDB database (Extended Data Fig. 5). In this comparison, the uncorrected flux calculations, as well as corrections from options 1 and 2, produce high fluxes at low concentrations, which lie outside the distribution of empirical observations. By contrast, the third correction better captures the relationship observed in the empirical observations. Thus, we used this final (third) correction when reporting the main results; however, we also note that large uncertainties remain when upscaling gas fluxes from river networks.

Finally, the approaches described above address only the diffusive $CH_4$ emissions, but $CH_4$ can also be emitted directly through ebullitive fluxes. However, ebullition measurements are scarce and more variable, making it hard to develop a robust estimate[3]. To provide an assessment of ebullitive $CH_4$ emissions from running waters, we explored patterns of $CH_4$ ebullition rates available in GRiMeDB. We assessed the magnitude and relationship between ebullitive and diffusive $CH_4$ fluxes by selecting observations from studies that contained both estimates. Those observations were filtered by excluding negligible and uncertain fluxes that were below 0.0001 mmol m$^{-2}$ per day, as well as excluding observations in which the $k_{600}$ was modelled using hydraulic relationships, which were more uncertain for local conditions and could substantially bias diffusive $CH_4$ estimates. We used the linear relationship between diffusive and ebullitive fluxes (in log-space; Extended Data Fig. 7) to preliminarily quantify uncertainty in ebullition. To do this, we used the regression model to represent the overall uncertainty: this included the uncertainty not only in the diffusive estimate itself but also in the relationship between diffusive and ebullitive fluxes derived from the prediction interval of the regression. Specifically, we calculated the prediction interval of the 2.5th and 97.5th percentiles of the diffusive estimate from the Monte Carlo simulation. We then selected the lower bound of the prediction interval of the 2.5th percentile and the higher bound of the 97.5th percentile as the overall uncertainty interval of the ebullitive estimate.

## Validation of modelled fluxes

Modelled diffusive fluxes were validated with directly measured $CH_4$ fluxes available in GRiMeDB. GRiMeDB includes about 7,300 diffusive flux measurements, and we selected observations that overlapped the river reaches present in GRADES (distance from the sampling site to the river reach is below 500 m) and compared the measured and modelled fluxes. For this exercise, we excluded the unclear methods used for $k_{600}$ estimation ($k$ method in GRiMeDB indicated as 'other' or 'not determined'). We also aggregated observations for a given reach of GRADES using mean values if multiple empirical observations were available, and we matched the monthly estimate to the month of the sampling date. This analysis (Extended Data Fig. 9a) shows that the modelled flux estimates fall within a similar magnitude as measured fluxes but were less variable and weakly related. Specifically, measured fluxes spanned seven orders of magnitude, whereas modelled fluxes were more constrained (two orders of magnitude). But both the modelled and measured fluxes are not entirely comparable. On the one hand, measured fluxes are often performed using chambers that have a small footprint (<10 m$^2$) or using hydraulic equations with measured local slope and velocity, both of which capture local hydrological processes on a given day. On the other hand, modelled fluxes rely on modelled average monthly discharge as well as river slope along a long reach (4–6 km) to obtain a $k_{600}$, together with the modelled $CH_4$

concentrations to calculate the flux. A fairer comparison would be to select occasions when the $k_{600}$ is similar for a given reach. When we compare measured and modelled fluxes for a given site for which the $k_{600}$ are comparable (modelled $k_{600}$ is between 0.5 and 1.5 times the measured $k_{600}$; Extended Data Fig. 9b, black points), the relationship between the two flux estimates is evident ($R^2 = 0.63$). The slope of this regression equation is 0.51 (Extended Data Fig. 9b, equation in the panel), indicating that the model overestimates fluxes at low values and underestimates fluxes at high values, but with a strong and notable relationship.

## Software used for the analysis

All data analysis, geographic information system processing, statistics and visualization were done using the R statistical software[52] (v.4.1.1). Packages used were dplyr (v.1.0.7) for data wrangling[53], ggplot2 (v.3.3.5) for visualization[54], lubridate (v.1.7.10) for temporal data[55], corr (v.0.4.3) to assess correlations in the data[56], ggtext (v.0.1.1) for labelling figures[57], ggpubr (v.0.4.0)[58] and patchwork (v.1.1.1)[59] for composing multipaneled figures, sf (v.1.0.3) for spatial analysis of vector data[60], terra (v.1.4.11) for spatial analysis of raster data[61] and rnaturalearth (v.0.1.0) for global base layers of rivers and oceans[62].

## Data availability

Global gridded and monthly maps of riverine $CH_4$ concentrations and emissions are available in the Zenodo repository (https://doi.org/10.5281/zenodo.7733577). Raw data to reproduce the analysis are available in the Zenodo repository (https://doi.org/10.5281/zenodo.7733604).

## Code availability

The code to reproduce the analysis, results and figures is found in GitHub (https://github.com/rocher-ros/RiverMethaneFlux).

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

**Acknowledgements** We thank S. Oliver, J. Rosentreter and E. Stets for providing insights for this project. The Swedish Research Council provided funding to G.R.-R. (grant no. 2021-06667), S.L. was funded by the National Key Research and Development Program of China (grant no. 2021YFC3200401) and E.H.S. was supported by the United States National Science Foundation (award DEB-2025982, NTL LTER). Any use of trade, firm or product names is for descriptive purposes only and does not imply endorsement by the US government.

**Author contributions** G.R.-R. conceived the study, performed the analysis and co-wrote the paper. E.H.S. conceived the study, compiled the database and edited the paper. L.C.L. aided in the analysis and compilation of the database and edited the paper. N.J.C. aided in the analysis and compilation of the database and edited the paper. P.A.R. conceived the study, provided insights and edited the paper. S.L. provided the code and materials for the analysis and edited the paper. G.A. aided in the modelling and edited the paper. R.A.S. conceived the study, provided insights, aided in the compilation of the database and co-wrote the paper.

**Funding** Open access funding provided by Swedish University of Agricultural Sciences.

**Competing interests** The authors declare no competing interests.

**Additional information**
**Correspondence and requests for materials** should be addressed to Gerard Rocher-Ros.

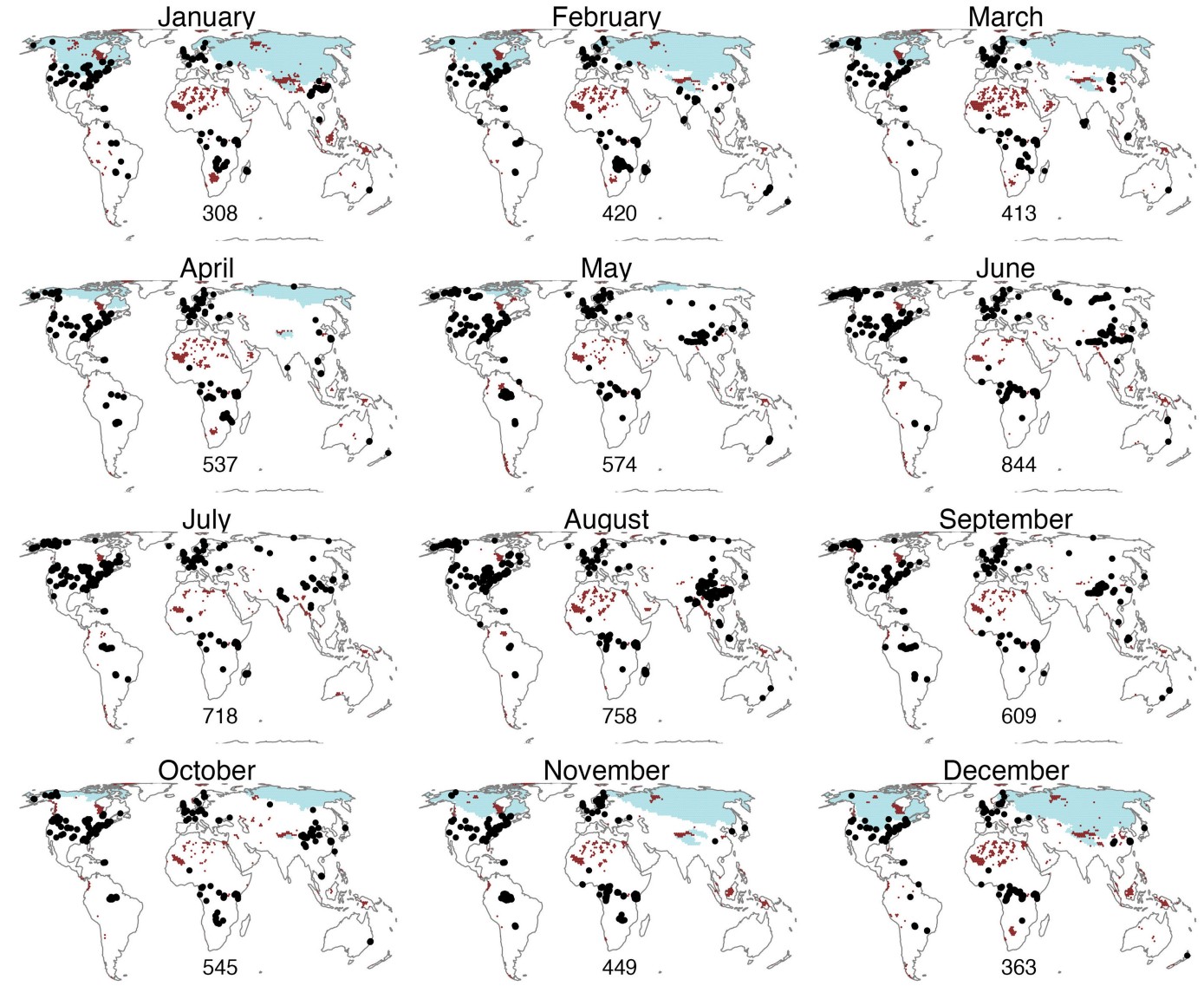

**Extended Data Fig. 1 | Observations in GRiMeDB[13] and their spatial representativeness.** Map of the globle showing the location of methane concentration observations in the database for each month (black points). The number in each panel denotates the total number of observations available each month, after aggregating temporal data for each subcatchment in GRADES. Light blue shows the snow or ice cover, and red polygons represent areas where the monthly model is extrapolating predictions (See section *"Random forest modelling"* in the main text for further explanation).

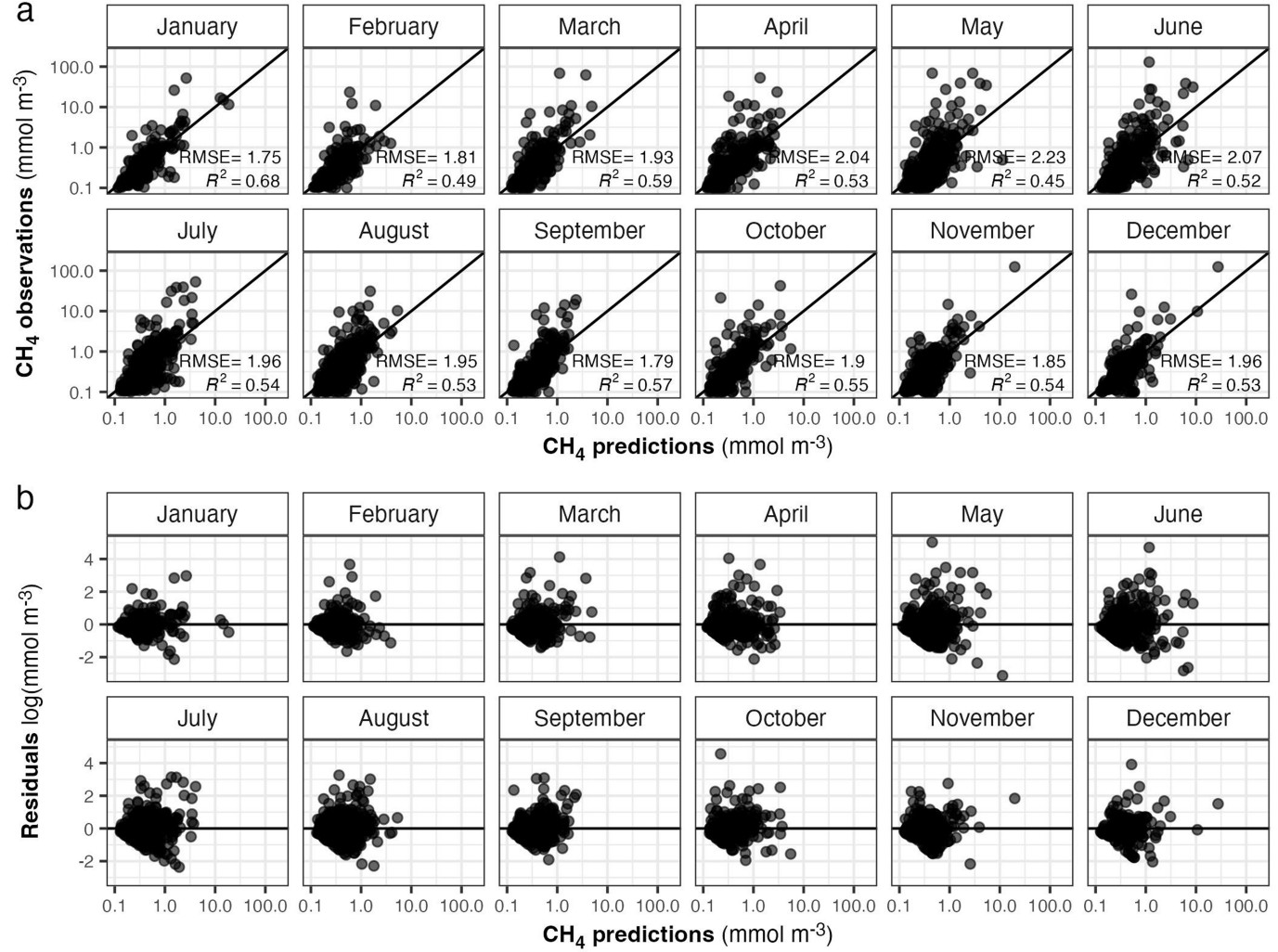

**Extended Data Fig. 2 | Performance of the random forest model. (a)** Predicted versus observed methane (CH₄) concentrations of the test dataset for each month, with the black solid line showing the 1:1 line. Inside the plot is shown the $R^2$ of the linear regression and the root mean square error (RMSE). **(b)** Residuals of the random forest model for predicted methane (CH₄) concentrations of the test dataset for each month. The black solid line is the x axis for y = 0.

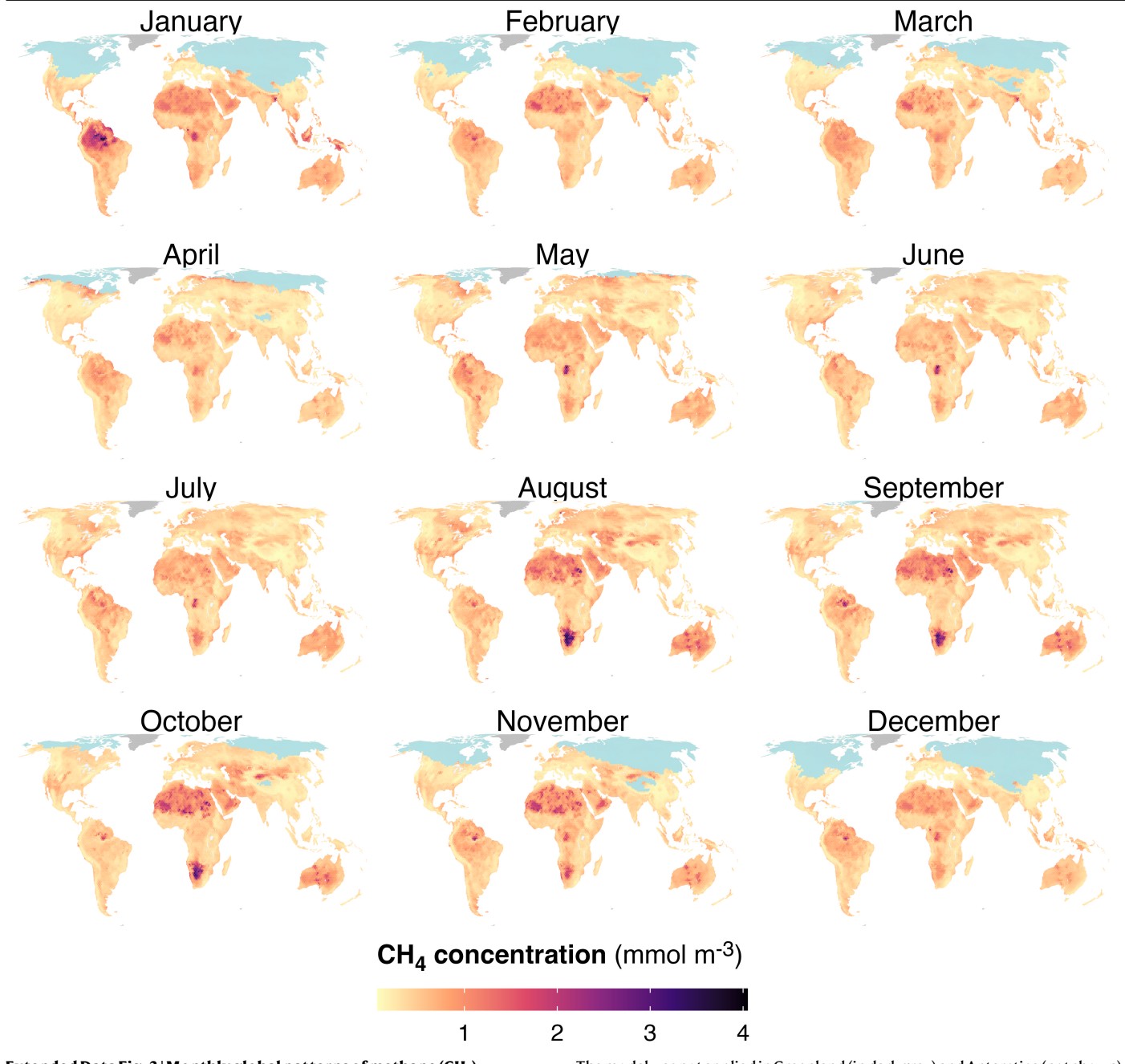

**Extended Data Fig. 3 | Monthly global patterns of methane (CH₄) concentrations.** Predicted CH₄ concentrations for the globe, using a random forest model for each month. Light blue areas indicate snow or ice cover. The model was not applied in Greenland (in dark grey) and Antarctica (not shown) due to lack of data coverage for many predictors.

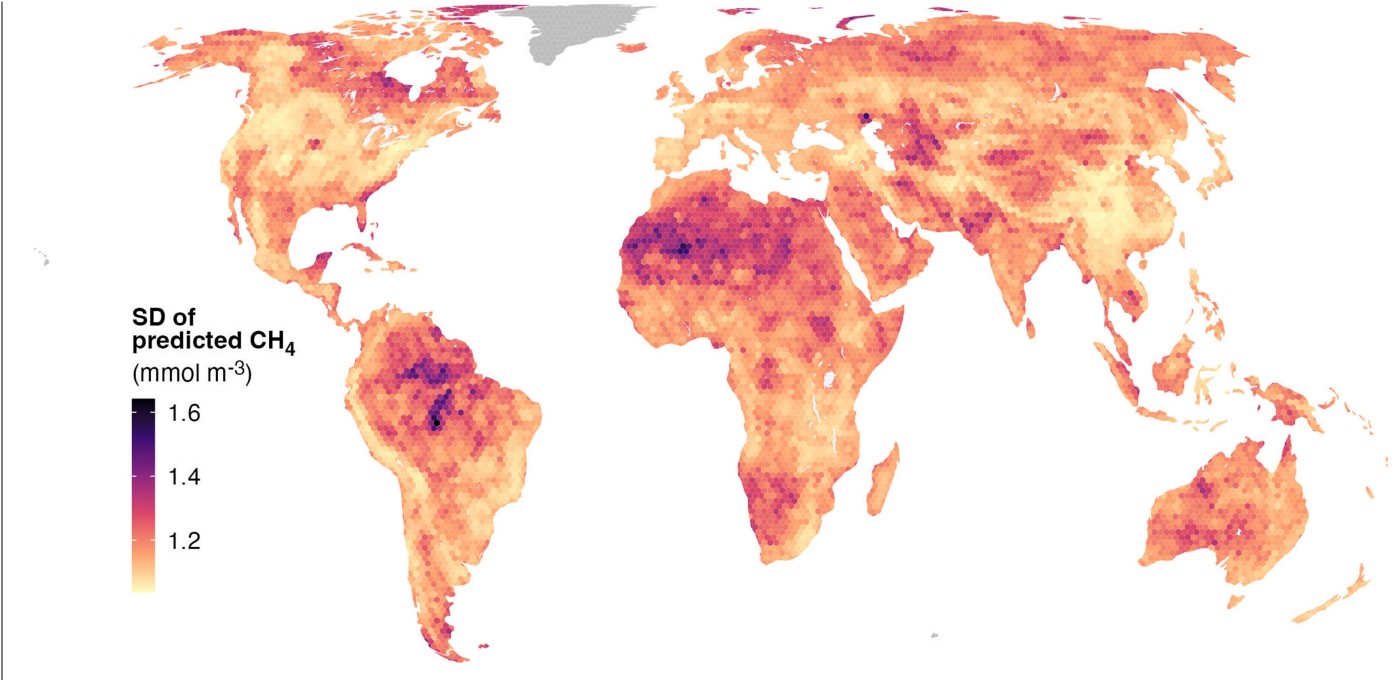

**Extended Data Fig. 4 | Uncertainty in river methane (CH₄) prediction.** Map of the standard deviation (SD) of the modelled CH₄ concentrations. The SD was obtained for each month and presented as a yearly average for this map. The model was not applied in Greenland (shown in dark grey) and Antarctica (not shown).

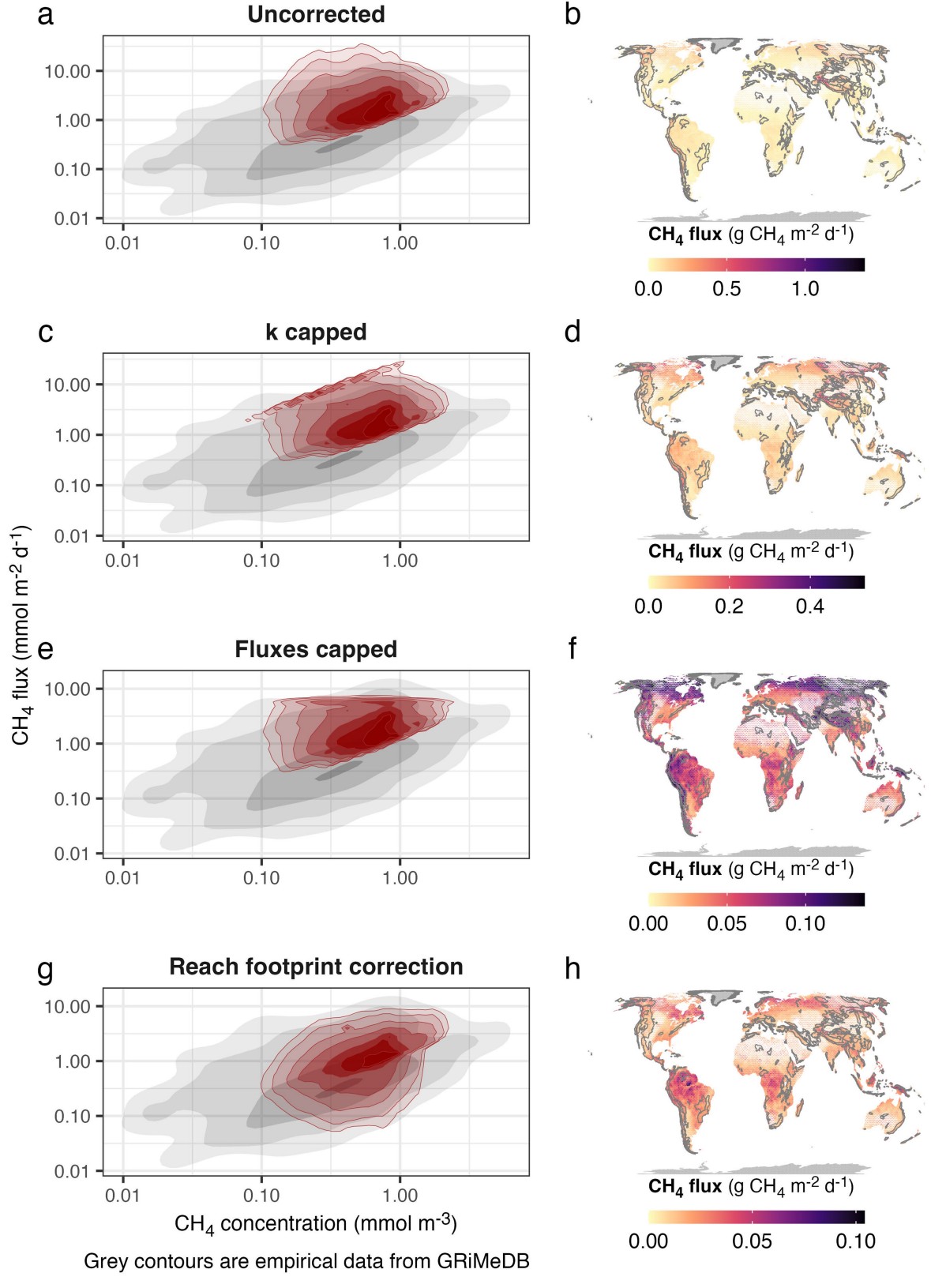

**Extended Data Fig. 5** | See next page for caption.

**Extended Data Fig. 5 | The effect of different correction methods for modeling methane (CH₄) concentration - flux relationships.** Uncorrected model estimates result in a large number of flux observations which are larger than observations reported in the empirical dataset. Plots in the left column provide a comparison between modelled and actual (observed) flux-concentration relationships. Red contours represent the density of modelled fluxes and concentrations, while grey contours represent the density of empirical observations available in GRiMeDB[13] (n = 4,052), with the lowest contours containing 95% of values. The maps in the right column illustrate modelled fluxes, with mountain areas highlighted as light grey, hollow polygons (from ref. 63); note the different scales for each row. Ideally, the modelled and empirical obervations of the concentration - flux relationship should overlap, but do not, as seen in (**a**), leading us to explore multiple corrections. Capping gas transfer velocity (*k*) (**c**, **d**) at 35 m per day does little to correct this artefact, nor does capping flux estimates above 2 standard deviations of the global population (**e**,**f**). In contrast, the river reach footprint correction avoids particularly the high fluxes at low concentrations and better represents the distribution of empirical observations (**g**, **h**). See methods section "*Uncertainty and refinement of the estimate*" for a detailed discussion of this issue and the approach selected.

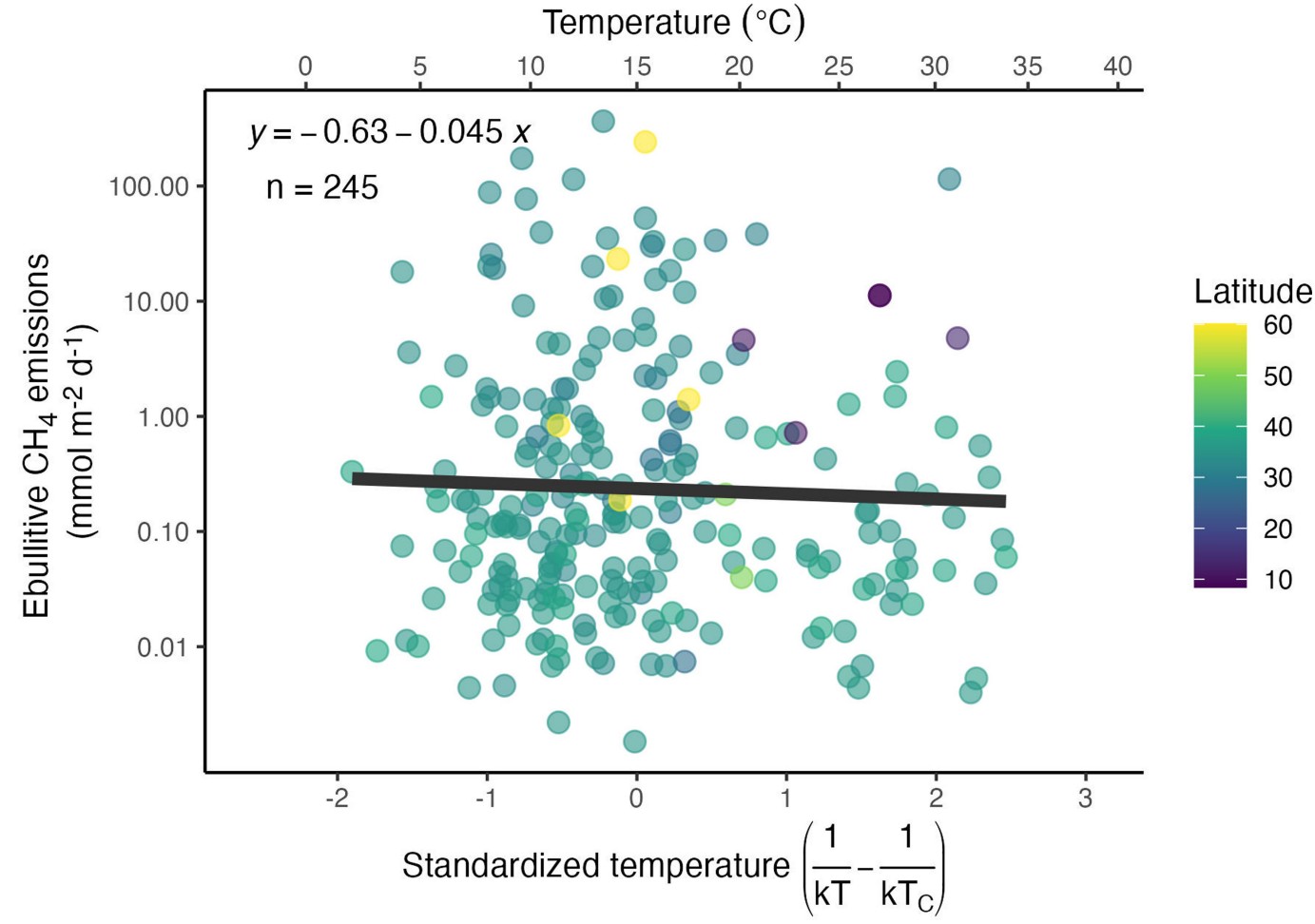

**Extended Data Fig. 6 | Temperature relationship of ebullitive methane (CH₄) fluxes.** Relationship between ebullitive CH₄ fluxes and temperature in GRiMeDB[13]. Here $k$ is the Boltzmann constant, T is water temperature in Kelvin, and $T_c$ is 15 °C (average water temperature in GRiMeDB). The total number of observations and model fit are shown in the figure, although the model is non-significant.

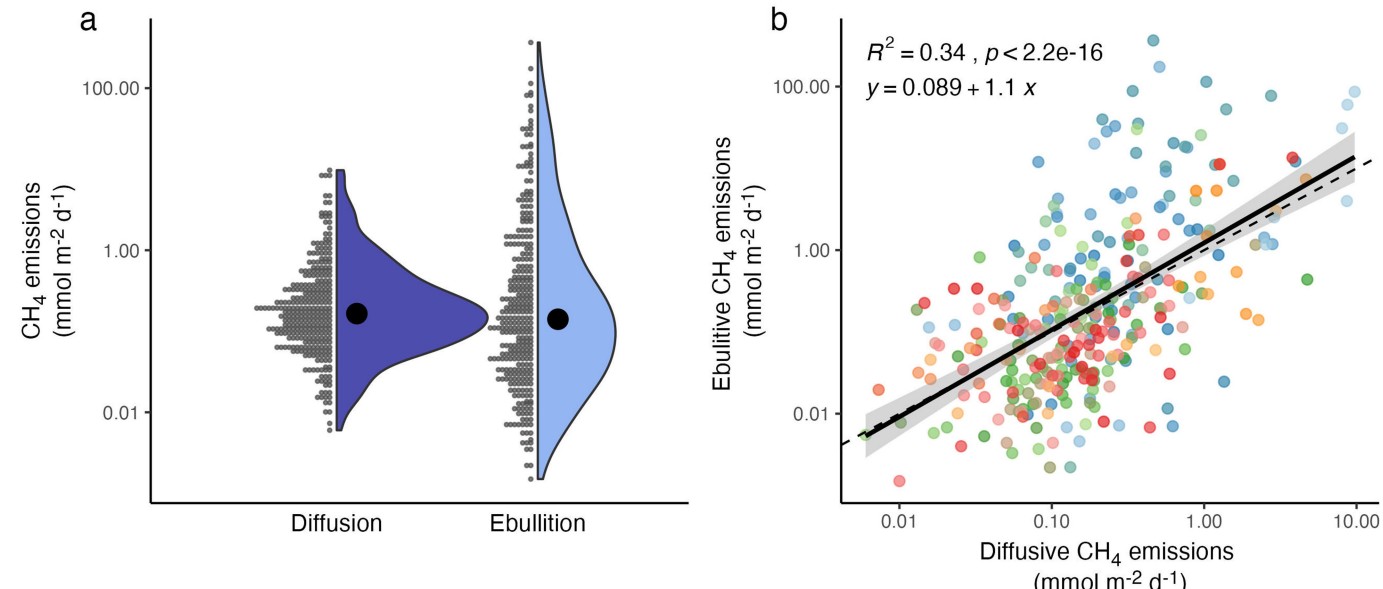

**Extended Data Fig. 7 | Assessment of ebullitive methane (CH₄) emissions.**
Simultaneous observations of diffusive and ebullitive emissions from
GRiMeDB[13] were measured simultaneously. (**a**) illustrates the overall
magnitude of the emissions, with the density plots (dark and light blue)
showing the distribution and dot plots showing each observation. The large
black dot represents the median value (n = 296). (**b**) shows the relationship
between diffusive and ebullitive emissions, where each site with more than one
measurement has a unique categorical color (number of sites = 93). The solid
line is a linear model fit (model fit and statistics reported in the upper left
corner of the plot) with the shaded polygon representing the 95% prediction
interval, and the dashed line represents the 1:1 ratio.

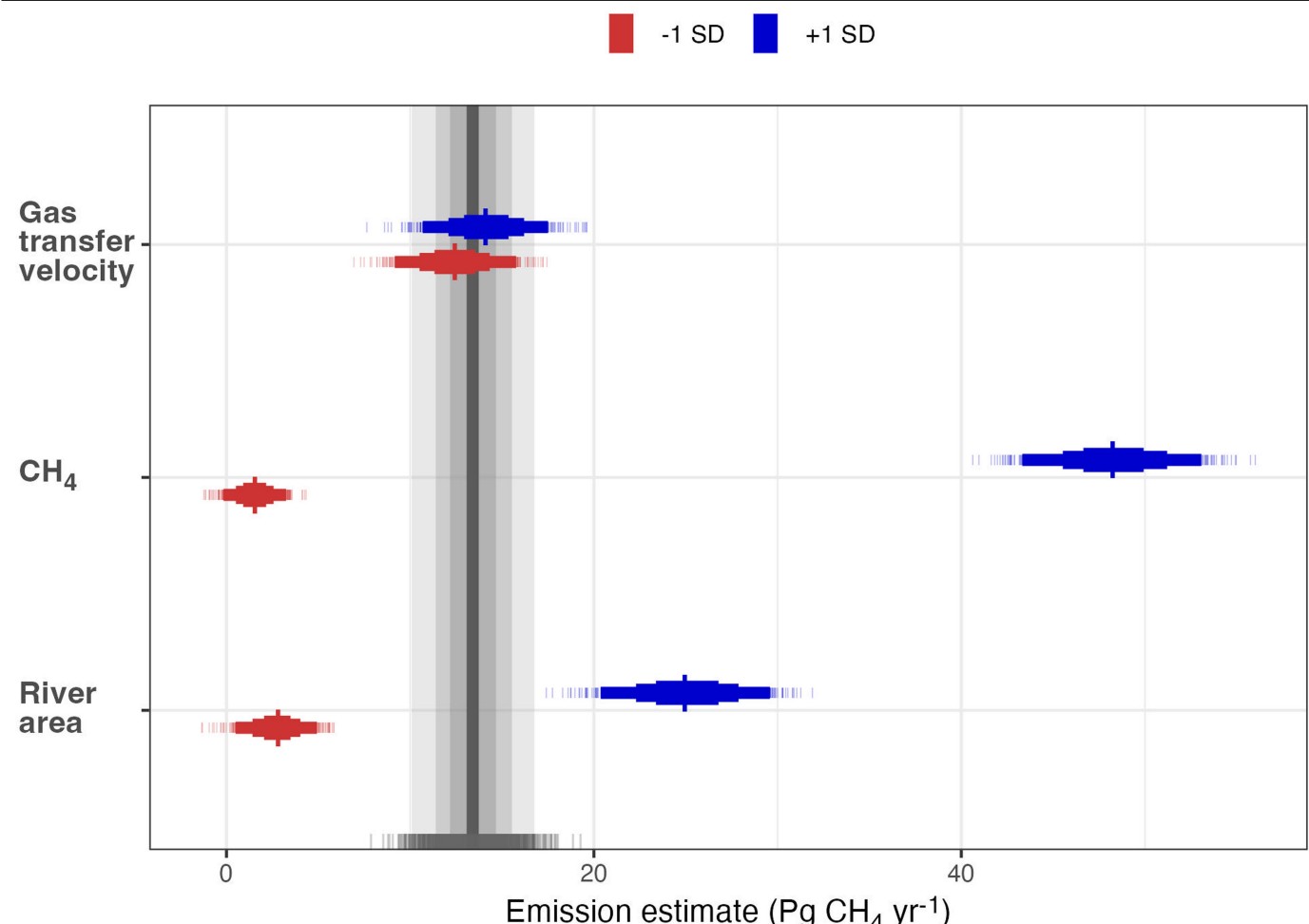

**Extended Data Fig. 8 | Sensitivity analysis of the global methane (CH₄) diffusive estimate.** The dark grey vertical line illustrates the mean estimate of global emissions, with lighter shades representing the 50, 75 and 95 % quantiles of the Monte Carlo simulation. The sensitivity analysis was performed for each flux parameter (gas transfer velocity, CH₄ concentration, river area) by increasing (blue) or decreasing (red) a parameter by 1 SD, then re-running the Monte Carlo simulation. Vertical coloured central lines show the mean values, with the rectangles decreasing in size showing the 50, 75 and 95 % percentiles. Thin vertical lines show the individual replicates of the Monte Carlo simulation for each experiment as well as for the main estimate (grey, bottom). Note that the uncertainty of the Monte Carlo simulation is highly sensitive to the uncertainty considered for each parameter, and thus it may change if other models are used.

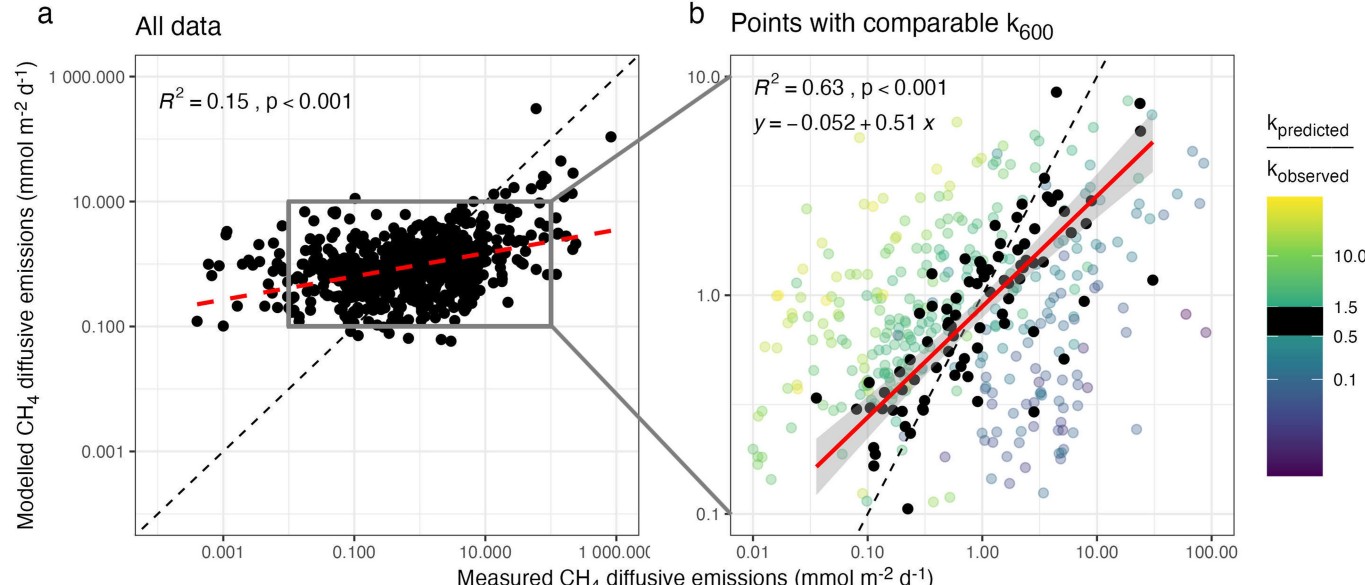

**Extended Data Fig. 9 | Comparison of modelled versus directly measured diffusive methane (CH₄) emissions.** (**a**) Modelled diffusive CH₄ fluxes in GRADES reaches where an empirical measurement is available in GRiMeDB. Measured fluxes span seven orders of magnitude while modelled fluxes are more constrained (2 orders of magnitude). One potential source of the discrepancy is the difference in gas transfer velocities (*k*) between the measured and predicted values, given that the vast majority of flux observations in GRiMeDB are measurements from a single day in a relatively short reach, while the modelled fluxes use monthly modelled discharge averages along a long river reach (4–6 km). When selecting pairs of observations with comparable *k* values, indicating that the hydrological conditions between the modelled and observed value are similar (predicted *k* is between 0.5 and 1.5 times the measured *k*; black points in (**b**)), the relationship between modelled and measured fluxes is evident. Dashed line in both panels is the 1:1 line, and note the change in the axis in panel (**b**).