## [Peer Review File · Nature]

Manuscript Title: Global methane emissions from rivers and streams

Reviewer Comments & Author Rebuttals

Reviewer Reports on the Initial Version:

Referees' comments:

Referee #1 (Remarks to the Author):

This manuscript is addressing an important research area of understanding global methane (CH₄) emissions. Overall, the manuscript is written clearly. Please find below my comments which are not necessarily ordered chronically or by important.

1. Data: The authors described the datasets used to generate Methane concentration (the response) and predictor variables. Many of the data seem to be at different spatial or temporal scale, and thus aggregation or averaging is used. The ecological fallacy is widely discussed when analyses on grouped/aggregated data could lead to different conclusions from the individual level data. The authors need to discuss this because it seems aggregation is used, and the model they used is mainly used to describe the relationship or association between the response and predictor variables.
2. If this study focuses more on making predictions of methane concentration, the R² or RSM from the random forest modelling seems to be moderate. I wonder if the authors did benchmark to demonstrate this would be the best to get. Have the authors tried other methods such as models emphasizing more on predicting (but maybe more difficult to explain)? Random forest is a method widely used and developed decades ago. It does possess nice properties regarding model interpretability (based on decision trees) but many latest methods in deep learning can provide improved predictions.
3. The authors mentioned that uncertainty was quantified by performing a bootstrapping procedure (Hoogen et al, 2021). I wonder if the authors can show results of the empirical coverage of uncertainty intervals in validation studies. For example, a 95% confidence or credible interval, if model assumptions are correct, empirically should have coverage probability 0.95.

Referee #2 (Remarks to the Author):

The work of Gerard Rocher-Ros and co-authors is based on the analysis of a large CH₄ data-set from which they analyze the patterns with a lot of spatial data (some of which is redundant and not independent) and then they compute the CH₄ emissions based on the river surface area the gas transfer velocity across the global river network recently reported by Liu et al. (2022). The data-set itself is a marginal improvement of the original data-base reported by Stanley et al. (2016). The general analysis of the drivers of CH₄ as function of soil and catchment properties is also a marginal improvement compared to the work of Stanley et al. (2016).

The authors try to build a case that the apparent temperature sensitivity of diffusive emissions from rivers is markedly lower compared to other systems (lakes, rice paddies, wetlands), and that this should also reveal some sort of disagreement with the universal temperature dependence of CH₄ production.

The fact that CH₄ emissions are different in lentic and lotic systems is not exactly a surprise, and

the low relation between emissions and temperature in rivers is probably related to the fact that there are confounding processes such as gas transfer velocity and river size that change with elevation, hence also temperature.

The fact that emissions and methanogenesis are two different processes and show a different relation to temperature should not come as a major surprise, and is not exactly a conceptual revolution.

Most of the main conclusions of this work on the drivers of CH₄ and the impact of human alterations are not new and redundant with the paper of Stanley et al. (2016). For the punch-line of the ms, the authors artificially created some sort of discrepancy in emission-temperature relations compare to methanogenesis and other aquatic systems. This "discrepancy" is not surprising since it is obvious that if you compare very different things, chances are they are, well, very different.

The results might be of interest to the community working on riverine methane, but should be of very limited interest to the wider community.

L 60 : More complex mathematical approaches do not necessarily allow to reduce « massive uncertainty », in some case the contrary occurs (Puy et al. 2022), and more complex mathematical approaches tend to produce more uncertain estimates.

L 66 : the new data-base is extremely similar to the one previously reported by Stanley et al. (2016).

L 76 : A lot of these factors are not independent.

Ground-water depth usually follows river slope and catchment slope.

Soil respiration follows primary production that is a function of precipitation and temperature.

So there is a huge redundancy in the factors used in the analysis.

L81-84: part of these differences could be explained by river size rather than differences in climate or biome. The high values in Arctic and boreal biomes could be due to the fact that extremely small systems were sampled.

L 141: I do not see an "apparent disagreement with the universal temperature dependence of CH₄ production". The fact that the delivery of CH₄ to rivers can be high in Arctic and boreal biomes because of very large stores of soil organic matter does not contradict temperature dependence of CH₄ production.

L 146 : Well, yes, "apparent temperature sensitivity of diffusive emissions from rivers is markedly lower" because you are looking at emissions and not methanogenesis. As these are two very different things and so unsurprisingly they show a different dependency to temperature.

The diffusive CH₄ emissions should change with river size, so there might be a difference in the emission-temperature relation between headwater streams and large lowland rivers.

Collectively, the effect of elevation and size probably blurs the emission-temperature relationships.

L 151 : This explanation is simplistic and probably incorrect. There should be a correlation between temperature and elevation, and a relation between elevation and gas transfer velocity.

L 180 : The impact of human alterations on CH₄ emissions is not new and was already given by Stanley et al. (2016).

Puy et al. (2022) Models with higher effective dimensions tend to produce more uncertain estimates, Science Advances, Vol 8, Issue 42, DOI: 10.1126/sciadv.abn9450

Referee #3 (Remarks to the Author):

Overall Review

This manuscript seeks to provide the most comprehensive estimate to date of global emissions of methane from rivers and streams worldwide. This estimate is important because most running waters are, at some point, supersaturated in CH₄. As such, they are an important and poorly quantified contributor of methane to the global atmosphere. If we want to understand human impacts on the global water cycle, we cannot ignore rivers and streams. The manuscript leverages several large databases in order to perform its estimate. Chief among these is GRIMEDB, a global database of river and stream methane concentration and emission values compiled from a large number of in situ studies. These are linked to the previously published GRADES database of global rivers, which includes estimates of discharge for rivers with basins larger than 25 km². The paper provides monthly estimates of methane concentration over the globe using a random forest modeling approach. It also examines possible controls on methane concentration by examining the partial dependence plots from the random forest analysis.

From my perspective, this is an interesting and important paper that deserves to be published in a widely read journal like Nature. The question it addresses is very important to understanding of global methane fluxes, which have been a source of considerable disagreement within the scientific community. I am convinced by the authors' arguments that this is the best estimate to date of CH₄ from rivers and streams. Moreover, the paper is very, very well written. My reviews ordinarily end with a sequence of comments about writing and figure clarity, but for this paper I have almost no comments in that vein at all. I do, however, have one overarching concern that I think the authors should address both to make their paper better and to spur further research in this area. At the moment, the handling of uncertainty in the manuscript is quite uneven. The consideration of error from the concentration analysis is relatively straightforward, as the random forest analysis provides uncertainty values. While concentration is important, however, the methane flux is the really important variable and what makes this paper deserving of publication in Nature.

In essence, we need to know four things to get methane flux: concentration, gas transfer velocity, and river/stream surface area produce evasion, and this is added to the flux from ebullition. The validation of methane diffusive flux comes in figure S10, and the model is just okay. It produces an r^2 of 0.32, and the best fit line deviates substantially from the 1-to-1 line, with high methane flux values substantially underestimated and low values substantially overestimated. The overall flux estimate presented in the abstract does not contain error bars at all, perhaps because the flux from ebullition also does not. There are a lot of additional potential sources of error here, and at the moment they are not particularly well laid out. I do not expect that the authors will be able to firmly quantify all potential sources of error, but I would like to see them comprehensively laid out. This will also require the authors to go back to the analysis from the Liu et al. (2022) paper that performs a similar analysis for CO₂, as the authors use the same method to quantify river surface area as that paper. I re-read that paper, and I have some concerns—for example, the method for estimating downstream hydraulic geometry relies on USGS stream gauges (See, for example, Allen and Pavelsky 2015: <https://agupubs.onlinelibrary.wiley.com/doi/full/10.1002/2014GL062764>). However, these gauges tend to be placed systematically at relatively narrow, stable locations on rivers, and extrapolating based on them may result in a substantial underestimate of river width in a DHG framework. This is just one example of uncertainty that is not, as far as I can tell, effectively quantified here. Note that I don't dispute the approach taken to calculating surface area as such—I'm not convinced there is an easier way to do so without attempting remote sensing of discharge on a monthly timescale, which could be accomplished using a global dataset of width variability like the one just published by Feng et al. (<https://agupubs.onlinelibrary.wiley.com/doi/full/10.1029/2021WR031712>). However, this would be a major undertaking and may be better left for a future endeavor. The main point is to at least

quantitatively or qualitatively address the magnitude of this source of uncertainty. There are certainly others related to surface area and gas transfer velocity.

What I would like to see is a table that provides an accounting of all sources of error that the authors can identify. For those that can be quantified, I would like to see that quantification performed and included in the table (e.g. the concentration uncertainty from the random forest model). For those that cannot be quantified, I would like to see at least a best effort to estimate how large the impact on CH₄ flux uncertainty is (1%? 10%? 100%?). Note that this table would also serve as an excellent spur to further progress in the future. If we have an excellent accounting of the important sources of uncertainty clearly laid out, that will provide a spur for the community to provide further improvements.

Aside from this one major comment, my few other comments are relatively minor.

Specific Comments

Figure 1: It's a little surprising to me to see large concentration and emission values in the middle of deserts (e.g. the Sahara and the Arabian desert). I recognize that the hexes have been scaled by runoff to try to mitigate this visual issue. I presume that the actual fluxes from these areas in the calculations are very small (as average discharge is very small), but I think the authors might want to reconsider how they present their data in these figures to better convey the relative balance of fluxes in different parts of the world.

Line 433: I would rewrite this as "~6900 diffusive flux measurements." "observations of" seems redundant with "measurements."

Line 435: Should be "reach," not "each."

Line 446: Should be "rely" not "relay."

Author Rebuttals to Initial Comments:

Referees' comments:

Referee #1 (Remarks to the Author):

This manuscript is addressing an important research area of understanding global methane (CH₄) emissions. Overall, the manuscript is written clearly. Please find below my comments which are not necessarily ordered chronically or by important.

1. Data: The authors described the datasets used to generate Methane concentration (the response) and predictor variables. Many of the data seem to be at different spatial or temporal scale, and thus aggregation or averaging is used. The ecological fallacy is widely discussed when analyses on grouped/aggregated data could lead to different conclusions from the individual level data. The authors need to discuss this because it seems aggregation is used, and the model they used is mainly used to describe the relationship or association between the response and predictor variables.

We agree that data aggregation is a key step that can affect the interpretation of the results, but this step is needed to go from local scale measurements to larger scale assessments of drivers. In our analysis, there are multiple steps where data are aggregated. First, spatial predictors, usually at a ~1-50 km resolution (see Table S1), are averaged for the subcatchment of each ~8-km river reach. Second, for all river reaches that have more than one methane observation in a given month (43% of the river reaches), we have used average values. Also relevant to this, we made a set of decisions to avoid potentially biased observations that are likely not captured by the spatial predictors; for example, we removed observations from ditches, canals, fracking-affected sites, downstream of dams, downstream of point sources and downstream of permafrost thaw slumps, which could have a distinct imprint on CH₄ dynamics that would not be captured at broader spatial scales by the predictors used. All those procedures are detailed in the methods (lines 270-278), which have been updated in response to the reviewer's comment.

Despite these methodological aspects, the largest issue when aggregating data is likely the fine-scale variability of methane concentrations in rivers. Some extensive spatial surveys (Crawford et al. 2019, Lupon et al. 2019) highlight the extreme spatial variability found in methane concentrations, even at the scale of 100 m stream reaches. Capturing this spatial variability in global models remains elusive until more detailed and spatially resolved hydrological datasets exist, likely at a scale < ~500 m. In this work, we are using river CH₄ measurements and using subcatchment landscape properties to explain riverine CH₄ patterns, and thus there is a gap in the spatial scales used. We discuss this at lines 116-120 of the main text, as a likely source of the unexplained model variance. When aggregating data, we have used median values instead of the mean to decrease the importance of highly skewed values. We have clarified this text in the methods and added some details regarding aggregation biases in the lines 282-286.

2. If this study focuses more on making predictions of methane concentration, the R² or RSM from the random forest modelling seems to be moderate. I wonder if the authors did benchmark to demonstrate this would be the best to get. Have the authors tried other methods such as models emphasizing more on predicting (but maybe more difficult to explain)? Random forest is a method widely used and developed decades ago. It does possess nice properties regarding model interpretability (based on decision trees) but many latest methods in deep learning can provide improved predictions.

We used random forest models as their use is widespread in the geosciences - because they are one of the most powerful algorithms for spatial prediction, and as the reviewer indicates, because their outputs are possible to interpret. The understanding of the spatial drivers of riverine methane emissions is an equal or even more important aim of this study, and therefore random forests seem the

most suitable tool for this task. However, for this revision, we explored whether other deep learning algorithms more oriented towards predictability could improve the results with this dataset. Accordingly, we used a gradient boosting algorithm “XGBoost” and a single-layer feed-forward neural network (Figure R1, see caption for implementation details) to explore this issue. Overall, we find that the performance of the three algorithms is similar, with a marginal improvement of the random forest model over the others. Interestingly, when the same models are implemented on the monthly datasets, the random forest model and the gradient boosting algorithms performs substantially better over the yearly dataset, while the neural network model underperforms markedly (See Table R1 for a monthly summary of key statistics).

Fig R1. Performance of the three machine-learning algorithms tested predicting methane concentrations, including the random forest model used in the original manuscript submission. All models were implemented using yearly averages of all values in each river reach. The x-axis are observations in the dataset withheld to the model, and in the y-axis the predicted values for each method, with the R and Root Mean Square Error in the plot. Gradient boosting was implemented using “XGBoost”, after a grid search to find the optimal combination of the parameters. The neural network is single-layer feed-forward, with 10 hidden units, implanted using “nnet”. More details on the hyperparameters used and R code is available in the script “2_model_selection.R” in github: (<https://github.com/rocher-ros/RiverMethaneFlux>)

Table R1. Model performance statistics (R^2 on top and [RMSE] in brackets) of the three algorithms tested, for each of the monthly datasets. Note that the tree-based algorithms performed generally better than on the yearly dataset, while the neural-network underperforms.

	Jan	Feb	Mar	Apr	May	Jun	Jul	Aug	Sep	Oct	Nov	Dec
Neural network	0.26 [2.5]	0.26 [2.54]	0.27 [2.3]	0.11 [2.72]	0.25 [2.43]	0.25 [2.58]	0.23 [2.28]	0.24 [2.48]	0.16 [2.25]	0.21 [2.32]	0.1 [2.68]	0.19 [2.71]
Random Forest	0.7 [1.73]	0.48 [1.81]	0.59 [1.93]	0.53 [2.04]	0.44 [2.24]	0.52 [2.07]	0.55 [1.95]	0.54 [1.94]	0.57 [1.79]	0.57 [1.88]	0.54 [1.85]	0.54 [1.96]
Gradient boosting	0.72 [1.65]	0.46 [1.80]	0.58 [1.82]	0.49 [2.15]	0.42 [2.1]	0.46 [2.28]	0.52 [1.87]	0.51 [2.03]	0.60 [1.88]	0.51 [1.88]	0.43 [2.09]	0.55 [2]

3. The authors mentioned that uncertainty was quantified by performing a bootstrapping procedure (Hoogen et al, 2021). I wonder if the authors can show results of the empirical coverage of uncertainty intervals in validation studies. For example, a 95% confidence or credible interval, if model assumptions are correct, empirically should have coverage probability 0.95.

We thank the reviewer for pointing to this assessment, which was missing. After point #2 (above) and comments of reviewer #3 (below), we have now re-tuned the model, redone the uncertainty analysis,

and performed a new sensitivity analysis. For the uncertainty in methane concentration estimates, instead of a rudimentary bootstrapping, we now quantify the variance of each estimate using the “infinitesimal jackknife” for random forests following Wager, Hastie and Efron (2014 in JMLR). In response to this specific comment, we have also quantified prediction intervals using quantile regression random forests. To do this, we assessed the coverage of the predictions with their associated uncertainty for the 20% withheld dataset, using the 75 and 95% prediction intervals, corresponding to the central 0.5, 0.75 and 0.95 observations from the quantile regression (see table R2). In this table, we provide the coverage of the 95% confidence interval of the mean value, quantified using the standard deviation obtained with the infinitesimal jackknife. The quantile regression prediction intervals appear to be conservative, with coverages higher than expected. The other uncertainty estimate that we use here with the infinitesimal jackknife shows a lower coverage, which could be due to uneven tails of the modelled distribution that are well captured by the quantile regression but not by a single value such as the SD. For simplicity purposes, and for the subsequent uncertainty quantifications in the upscaling procedures, we chose to report the SD obtained with the infinitesimal jackknife in the global maps of uncertainty and data products created in this study.

Table R2. Empirical coverage of the model, shown in percentage. For the 20%-withheld dataset, we assessed whether observations fell inside the prediction interval for a given modelled value. For example, for the 90% prediction interval estimated using quantile regression, 95% of the observations were inside the uncertainty ranges.

Empirical Coverage	Jan	Feb	Mar	Apr	May	Jun	Jul	Aug	Sep	Oct	Nov	Dec
90% prediction interval	95	92	95	92	94	94	92	94	92	95	96	92
75% prediction interval	80	76	84	77	84	81	83	82	78	83	83	78
95% confidence interval (mean $\pm 1.96SD$)	87	83	91	82	92	83	81	88	86	82	84	85

Referee #2 (Remarks to the Author):

The work of Gerard Rocher-Ros and co-authors is based on the analysis of a large CH₄ data-set from which they analyze the patterns with a lot of spatial data (some of which is redundant and not independent) and then they compute the CH₄ emissions based on the river surface area the gas transfer velocity across the global river network recently reported by Liu et al. (2022). The data-set itself is a marginal improvement of the original data-base reported by Stanley et al. (2016). The general analysis of the drivers of CH₄ as function of soil and catchment properties is also a marginal improvement compared to the work of Stanley et al. (2016).

Comparisons to Stanley et al. (2016) resurface a few times in the comments below, where we provide more specific responses. Overall, we disagree strongly that this study represents a marginal advance over that previous work. While the 2016 article presented a review and conceptual framework for understanding riverine methane dynamics, the dataset analyzed in that study is orders of magnitude smaller in terms of the overall number of observations, the global coverage of sites, and the typology of running-water systems represented. The prior data also lacked any temporal resolution. A useful example of this massive data expansion is provided by China: the dataset used in the 2016 paper included 4 sites, each with a single concentration value. Here, we draw on 1859 concentration observations from 468 sites from this country.

Next, while the 2016 paper includes some analyses addressing relationships between methane and other physical and chemical variables (mostly univariate correlations and non-parametric comparisons among land use/land cover categories), there is no formal spatial analysis comparable to what we

present in this study. Nor was any there an opportunity to address the temperature-emission relationships we investigate here with the data available at that time. Indeed, the major conclusion drawn from the analyses in the 2016 paper was that there was too much variance in the data to identify global scale predictors, and in fact, such predictors might not exist (p. 152 of Stanley et al. 2016). By contrast, this study, informed by a far richer data set and a far more sophisticated multivariate analysis, demonstrates the error of this prior conclusion and allowed us to generate the first spatially and temporally explicit understanding of patterns and drivers on river methane concentrations and emissions at the global scale.

The authors try to build a case that the apparent temperature sensitivity of diffusive emissions from rivers is markedly lower compared to other systems (lakes, rice paddies, wetlands), and that this should also reveal some sort of disagreement with the universal temperature dependence of CH₄ production. The fact that CH₄ emissions are different in lentic and lotic systems is not exactly a surprise, and the low relation between emissions and temperature in rivers is probably related to the fact that there are confounding processes such as gas transfer velocity and river size that change with elevation, hence also temperature.

This is an important comment that we have worked to clarify in the manuscript (see lines 146-174). First, it is not our intent to suggest that the temperature sensitivity of methanogenesis is somehow different in streams and rivers from other ecosystems. Instead, our point is that, unlike other aquatic systems (e.g., lakes and wetlands), methane *emissions* from running waters are not strongly temperature sensitive at a global scale. The overall lack of thermal influence can be seen within individual sites over time, as well as across sites spanning steep climate gradients, and is further reflected in global pattern where the highest daily emission rates are observed in both tropical and high latitude settings (i.e., in both the warmest and coldest biomes).

While the reviewer may not find this result surprising, the idea of strong temperature-dependence of methane emissions from aquatic ecosystems pervades the literature without being rigorously evaluated. First and foremost, the field has been strongly influenced by the seminal paper from Yvon-Durocher et al. (2014, in *Nature*), which suggests that methane emissions are strongly (and similarly) sensitive to temperature across all aquatic ecosystems, including rivers. Indeed, one major point from that paper is that the temperature-dependence of ecosystem-scale evasion/emissions from aquatic ecosystems and wetlands closely matches the theoretical expectations from methanogenesis. Our results indicate that this is not the case for riverine ecosystems across the globe.

The influence of this thinking is also clear in more recent high-profile papers. As just one example, in a recent review of greenhouse emissions from running waters (Pilla et al. 2022, *Global Change Biology*), the authors explicitly state the expectation that warming will increase methane emissions from running waters through indirect effects of permafrost thaw at high latitudes, but primarily through the direct effects of aquatic metabolic activity, leaning almost entirely the Yvon-Durocher et al. 2014 paper as evidence for this claim. In this context, the authors draw no distinctions between streams, rivers, and lakes (all will respond the same) - similar to the approach by Yvon-Durocher et al. 2014 where the temperature dependence of CH₄ emissions from lakes and rivers were analyzed jointly.

Overall, our results thus stand in clear contrast to a basic idea related to the control over riverine methane emissions that continues to be propagated in the field, as well as to what is commonly observed in lakes and wetlands, where emissions to the atmosphere can show strong thermal controls (as examples, see Jansen et al. 2020 (for lakes); Hopple et al. 2022 (for wetlands)). Once more, these findings do not mean that there is a different or unique temperature-dependence for methanogenesis in streams and rivers, but they do demonstrate that the internal (aquatic) production of CH₄ in running waters is less important than external (e.g., lateral) supplies of CH₄ when it comes to understanding and predicting riverine emission at broad scales. A key consequence of this result is that climate warming may not necessarily lead to elevated emissions from fluvial systems as is widely predicted. Instead, the consequences of climate change for riverine CH₄ emissions are far more likely to be

shaped by how warming and altered precipitation patterns influence methanogenesis elsewhere in the landscape and how this CH₄ is subsequently delivered to flowing waters.

In our responses below, we address several similar questions about the temperature dependence issue, including some linked to potentially confounding factors in our analysis. In this context, several of the factors suggested by the reviewer (e.g., river size, elevation, etc.) are in fact corrected for in our analysis, in part because we are able to focus the assessment of temperature-dependence within a large number (n = 223) of individual sites. We thus believe that our analysis of this new dataset provides important clarity on this question.

The fact that emissions and methanogenesis are two different processes and show a different relation to temperature should not come as a major surprise, and is not exactly a conceptual revolution.

See comment above. Briefly, we agree – yet as already discussed, this assumption has not been widely and rigorously tested for fluvial systems, and the view that temperature does influence emissions for these ecosystems is pervasive. Further, for other aquatic ecosystems and wetlands, temperature dependence of methane emissions can indeed be very similar to that of methanogenesis (as emphasized by Yvon-Durocher et al. 2014). Recognizing that this consistency does not hold true for running water ecosystems is critically important for understanding how methane emissions from these ecosystems may respond to global environmental change.

Most the main conclusions of this work on the drivers of CH₄ and the impact of human alterations are not new and redundant with the paper of Stanley et al. (2016). For the punch-line of the ms, the authors artificially created some sort of discrepancy in emission-temperature relations compare to methanogenesis and other aquatic systems. This “discrepancy” is not surprising since it is obvious that if you compare very different things, chances are they are, well, very different.

First, Stanley et al. (2106) assessed these patterns using a far smaller dataset that ironically excluded sites overtly affected by humans (e.g., downstream from reservoirs, point source inputs, canals and ditches). The 2016 paper did find evidence that sites in urban and agricultural areas tended to have higher average CH₄ values, but also that there was substantial variability among land use categories. As such, it provided more of a suggestion of human influence in the specific case of land use rather than a definitive demonstration across a range of human modifications. Here, we examine several typologies of anthropogenic influence and given the larger and more diverse dataset, differences reported here are far more apparent than in the prior exploratory analysis in Stanley et al. 2016.

Regarding temperature dependency, please refer to the text above. To reiterate here, we do not present any data describing temperature-methanogenesis relationships. However, the lack of temperature-dependence of methane emissions from running water is notably distinct from other freshwater systems and wetlands.

The results might be of interest to the community working on riverine methane, but should be of very limited interest to the wider community.

We hope that, after improvements to the manuscripts and the clarification provided here, the reviewer better appreciates our results in the broader context of the methane cycle. In our view, the general interest lies in presenting the first robust, and spatially and seasonally revolved estimate of a greenhouse gas flux that is globally relevant in magnitude. In doing so, we also reveal a global pattern that has no precedence in the literature and elevates a general message that climate and other environmental changes on land are likely to have major implications for riverine CH₄ fluxes globally. Finally, from a basic science perspective, our work challenges an assumption about the temperature dependence of methane emissions from the Earth's stream and rivers, which, based on our findings, appears to be an overly simplistic perspective – and one that is inconsistent with the open and dynamic nature of these ecosystems.

L 60: More complex mathematical approaches do not necessarily allow to reduce « massive uncertainty », in some case the contrary occurs (Puy et al. 2022), and more complex mathematical approaches tend to produce more uncertain estimates.

The reference provided by the reviewer discusses issues of increased uncertainty for mechanistic, process-based models. This problem is indeed a concern for many biogeochemical and hydrological models used in Earth System Models, but it is not relevant for the analyses used in our paper. The current lack of knowledge regarding the specific processes and controls of riverine methane emissions is so substantial that developing a process-based model that could suffer from the particular pitfalls noted above is not yet possible. However, in our manuscript, we provide the first step toward developing a robust process-based model by characterizing the main drivers of riverine methane emissions. For this, we use statistical machine learning models, which are fundamentally different from the ones treated in Puy et al. 2022. Random forest models are not affected by extra variables—quite the opposite: adding a new variable that may be the product of the interaction of two other variables already present in the model allows the trees to better classify values that otherwise would have a larger range (Biau and Scornet 2016). It is not until adding a larger number of superfluous variables that the performance can be affected, but here the variables are already limited.

L 66 : the new data-base is extremely similar to the one previously reported by Stanley et al. (2016).

This is simply not true (see figure below). A recent rapid increase in data availability (Fig. R2) means that the new methane database has >25 times more observations for concentration data (>24,000 observations vs. <1000) and 19 times more observations for methane emissions. Importantly, this new database also provides temporally-resolved observations for a large number of sites (e.g., 393 sites with more than 10 concentration observations and 224 sites with more than 10 flux observations). This is a massive increase in spatial data that allows us to, for the first time, to explore and reveal global patterns of riverine methane emissions. Also, as described above, the within-site data compilation further allows us to evaluate the temperature sensitivity of river emissions without confounding effects of biome or rivers size. To better comprehend the importance and improvements of the new database, we refer the reviewer to the accompanying data paper currently in review in ESSD and available as a preprint (<https://essd.copernicus.org/preprints/essd-2022-346/>)

Figure R2: Number of observations of river methane concentrations (orange) and fluxes (green). Vertical line is the year 2015, when the dataset used by Stanley et al. (2016) was published. Note that Stanley et al. had data aggregated by site, and thus the number of observations in the previous database is far lower than the intersect of the vertical line.

L 76 : A lot of these factors are not independent. Ground-water depth usually follows river slope and catchment slope. Soil respiration follows primary production that is a function of precipitation and temperature. So there is a huge redundancy in the factors used in the analysis.

It is true that some variables are correlated. However, we removed variables that were highly correlated and thus redundant, with a *pearson* $r > 0.9$. Furthermore, random forest models are not as affected by correlated factors as other models may be, and in fact it can improve overall model performance to include relevant variables that are interrelated. See comment and reference above about random forests for more details. Finally, regarding the interpretation of the results: we do not make claims linking methane concentrations or emissions to any single factor, but instead discuss categories (or classes) of factors that covary with response variables.

L81-84: part of these differences could be explained by river size rather than differences in climate or biome. The high values in in Arctic and boreal biomes could be due to the fact that extremely small systems were sampled.

The reviewer suggests that the relatively high CH₄ concentrations at high latitudes compared to lower latitudes is because at high latitudes, smaller river systems are preferentially sampled compared to temperate and tropical areas. This is not true (see figures R3 below). In the upper panel, we compare stream size in the database across latitudinal bands, and at both high and low latitudes the data structure is similar. In addition to this, we find no strong trend in methane concentrations with river size within those latitudinal bands (see lower panel). Collectively, these assessments indicate that the observed global pattern in concentration and emissions is in fact not an artifact of different sized-size ecosystems over- (or under-) represented in any region.

Figure R3: Relative distribution of sites across river Strahler order. In the upper panel, we separated sites across latitudinal bands as in Battin et al 2023 (*Nature*), where “High Latitude is > 60 degrees, “Low latitude” below 25 degrees, and “Mid latitude” sites in between. Latitudinal degrees are in WGS84 and absolute values (i.e., northern and southern hemispheres were combined). The lower panel show boxplots of methane concentrations across river order for each latitudinal band.

L 141: I do not see an “apparent disagreement with the universal temperature dependence of CH4 production”. The fact that the delivery of CH4 to rivers can be high in Arctic and boreal biomes because of very large stores of soil organic matter does not contradict temperature dependence of CH4 production.

See our general comment above re: temperature-dependence of methanogenesis versus emissions (and see edits to manuscript at lines 146-174). The key word in this passage is ‘*apparent.*’ In this paragraph, we draw attention to the contrast between temperature dependency of sediment methanogenesis, which in some cases (and for some aquatic ecosystems), translates to clear and quantitatively similar temperature dependence of emissions, which we do not observe for running waters at the global scale. Clearly, if temperature was the first-order global driver, we should see low emission rates in the boreal and Arctic biomes, but we do not. Indeed, we agree with the some of this

reviewer's other comments above and are making the same point here: there are other processes at play that swamp out the role of temperature on aquatic sediment methanogenesis at broad scales. Most notably, the globally high values we observe for boreal and Arctic sites can only be explained by the idea that concentrations and emissions in these landscapes are regulated by hydrological connections to organic-rich, anoxic soils – rather factors that regulate methanogenesis within aquatic sediments. We have attempted to clarify the wording to eliminate any confusion.

L 146 : Well, yes, “apparent temperature sensitivity of diffusive emissions from rivers is markedly lower” because you are looking at emissions and not methanogenesis. As these are two very different things and so unsurprisingly they show a different dependency to temperature.

Please see the general comment above re: temperature sensitivity and the response to the comment regarding L141. Again, others have similarly analyzed the temperature dependence of emissions from ecosystems and found it matched that of methanogenesis even though these are two very different processes. Although such an analysis has not previously been done uniquely in rivers, our earlier text provided examples to demonstrate the belief that emissions are temperature dependent in rivers is common in the literature.

The diffusive CH₄ emissions should change with river size, so there might be a difference in the emission-temperature relation between headwater streams and large lowland rivers. Collectively, the effect of elevation and size probably blurs the emission-temperature relationships.

Response: Again, we have focused on the within-site responses of emissions to temperature specifically to remove any effect of river size or elevation from the analysis (or any other spatial factor that may confound responses). Furthermore, in figure S11 we show that even within larger rivers (which may behave more as “closed systems”) it was still not possible to capture a general temperature dependence of sediment methanogenesis in surface water emissions.

Below, we further assess if there are any patterns in the temperature dependence within sites across stream orders (Figure R4 below). Although we acknowledge that there are more time series of methane flux from smaller systems in the database, there are several sites at Strahler orders 4-6, that allow for a tentative assessment. From this, we find no clear trend in the apparent activation energy of emissions across stream orders. To confirm that there is no effect of elevation, we have also plotted Em versus elevation. Again, with sites clustered at low elevations, but there is nonetheless no clear pattern suggesting artifact of stream elevation (Figure R4)

Figure R4: Apparent activation energy of river methane emissions, classified by river order and against elevation. Left plot shows each within-site estimate and boxplots per river order. Estimates were performed as detailed in the manuscript.

L 151 : This explanation is simplistic and probably incorrect. There should be a correlation between temperature and elevation, and a relation between elevation and gas transfer velocity.

We do not quite understand the reference to L151, which states: ‘We attribute these low EM estimates to the fundamentally ‘open’ nature of running waters, where external inputs account for a large fraction of C gases evaded to the atmosphere but also fuel aquatic metabolic processes through organic matter supply.’

Here we simply suggesting that the low apparent activation energy of river emissions is due to the disconnection between the methane emitted in rivers and its sources in the surrounding catchment (e.g., near stream soils, floodplains, etc.). Methanogenesis in these lateral sources may indeed be temperature sensitive, but this Em signal is not translated through to river emissions. The second part of the reviewers’ comment is correct, elevation and temperature and elevation and gas transfer velocity are related, but given that we examined within-site temperature dependence of methane emissions (and thus includes the effect of k) and that most of sites are at low elevations, we are unsure how to further address this comment.

L 180 : The impact of human alterations on CH4 emissions is not new and was already given by Stanley et al. (2016).

Response: We agree that this idea is not new and we do not claim otherwise. Yet, as stated above, we are able to use a far larger database to provide a more refined assessment of varied anthropogenic influences. This includes an estimate of the effects of these activities on emissions, which was not done in the Stanley et al. paper.

Puy et al. (2022) Models with higher effective dimensions tend to produce more uncertain estimates, Science Advances, Vol 8, Issue 42, DOI: 10.1126/sciadv.abn9450

Referee #3 (Remarks to the Author):

Overall Review

This manuscript seeks to provide the most comprehensive estimate to date of global emissions of methane from rivers and streams worldwide. This estimate is important because most running waters are, at some point, supersaturated in CH4. As such, they are an important and poorly quantified contributor of methane to the global atmosphere. If we want to understand human impacts on the global water cycle, we cannot ignore rivers and streams. The manuscript leverages several large databases in order to perform its estimate. Chief among these is GRIMEDB, a global database of river and stream methane concentration and emission values compiled from a large number of in situ studies. These are linked to the previously published GRADES database of global rivers, which includes estimates of discharge for rivers with basins larger than 25 km². The paper provides monthly estimates of methane concentration over the globe using a random forest modeling approach. It also examines possible controls on methane concentration by examining the partial dependence plots from the random forest analysis.

From my perspective, this is an interesting and important paper that deserves to be published in a widely read journal like Nature. The question it addresses is very important to understanding of global methane fluxes, which have been a source of considerable disagreement within the scientific community. I am convinced by the authors’ arguments that this is the best estimate to date of CH4 from rivers and streams. Moreover, the paper is very, very well written. My reviews ordinarily end

with a sequence of comments about writing and figure clarity, but for this paper I have almost no comments in that vein at all. I do, however, have one overarching concern that I think the authors should address both to make their paper better and to spur further research in this area. At the moment, the handling of uncertainty in the manuscript is quite uneven. The consideration of error from the concentration analysis is relatively straightforward, as the random forest analysis provides uncertainty values. While concentration is important, however, the methane flux is the really important variable and what makes this paper deserving of publication in Nature.

In essence, we need to know four things to get methane flux: concentration, gas transfer velocity, and river/stream surface area produce evasion, and this is added to the flux from ebullition. The validation of methane diffusive flux comes in figure S10, and the model is just okay. It produces an r^2 of 0.32, and the best fit line deviates substantially from the 1-to-1 line, with high methane flux values substantially underestimated and low values substantially overestimated. The overall flux estimate presented in the abstract does not contain error bars at all, perhaps because the flux from ebullition also does not. There are a lot of additional potential sources of error here, and at the moment they are not particularly well laid out. I do not expect that the authors will be able to firmly quantify all potential sources of error, but I would like to see them comprehensively laid out. This will also require the authors to go back to the analysis from the Liu et al. (2022) paper that performs a similar analysis for CO₂, as the authors use the same method to quantify river surface area as that paper. I re-read that paper, and I have some concerns—for example, the method for estimating downstream hydraulic geometry relies on USGS stream gauges (See, for example, Allen and Pavelsky 2015: <https://agupubs.onlinelibrary.wiley.com/doi/full/10.1002/2014GL062764>). However, these gauges tend to be placed systematically at relatively narrow, stable locations on rivers, and extrapolating based on them may result in a substantial underestimate of river width in a DHG framework. This is just one example of uncertainty that is not, as far as I can tell, effectively quantified here.

Note that I don't dispute the approach taken to calculating surface area as such—I'm not convinced there is an easier way to do so without attempting remote sensing of discharge on a monthly timescale, which could be accomplished using a global dataset of width variability like the one just published by Feng et al. (<https://agupubs.onlinelibrary.wiley.com/doi/full/10.1029/2021WR031712>). However, this would be a major undertaking and may be better left for a future endeavor. The main point is to at least quantitatively or qualitatively address the magnitude of this source of uncertainty. There are certainly others related to surface area and gas transfer velocity.

What I would like to see is a table that provides an accounting of all sources of error that the authors can identify. For those that can be quantified, I would like to see that quantification performed and included in the table (e.g. the concentration uncertainty from the random forest model). For those that cannot be quantified, I would like to see at least a best effort to estimate how large the impact on CH₄ flux uncertainty is (1%? 10%? 100%?). Note that this table would also serve as an excellent spur to further progress in the future. If we have an excellent accounting of the important sources of uncertainty clearly laid out, that will provide a spur for the community to provide further improvements.

We thank the author for the positive comments. We agree that refining the areal estimates of global river networks is a major source of uncertainty that would greatly refine the global estimate. As the reviewer points out, that is a major endeavor that will define global river biogeochemistry in the next decade, but is also out of our current reach.

In the original manuscript, we propagated uncertainty not only from the modelled concentrations but also k to estimate fluxes. In the revised version, we have redone the uncertainty analysis by:

1. Generating a better assessment of uncertainty of modelled CH₄ concentrations, which resulted in an increase in the variance of the modelled CH₄ concentrations. The previous bootstrapping procedure was quite rudimentary and did not include important components of model

uncertainty. The new procedure better captures this uncertainty, including estimates outside the training dataset. The empirical validation of this uncertainty is addressed in the responses to reviewer #1.

2. Instead of propagating uncertainty using basic rules of propagation, we have now used a Monte Carlo simulation to better resolve sources of variability and estimate the sensitivity of each parameter. We have also included uncertainty in river width, derived from the SD of the daily river discharge values in GRADES. From this, we can now quantify the effects of CH₄ concentration, k and river area on the global estimate, and thus inform where the future challenges remain in terms of quantifying river methane emissions. A summary of this uncertainty is in the SM but also in the figure below (Figure R5).
3. We have made a first attempt to assign uncertainty to the ebullition estimate. For this we have used the regression model in figure S13, which relates diffusive and ebullitive emissions from empirical observations. We report now the uncertainty from two sources here, the uncertainty in the global diffusive estimate and the uncertainty from the regression model shown in figure S13.

This change in the modelling and upscaling procedure has increased the overall uncertainty around the diffusive methane emission estimate, with a mean [95% confidence interval] of 13.5 Pg CH₄ [10.1 – 16.8]. Similarly, the ebullitive estimate is now 14.5 [6.6 - 22.9].

Figure R5: Sensitivity analysis of the global estimate by performing a monte carlo simulation on the k and CH₄ parameters. For each parameter, we have increased or decreased 1 standard deviation the values for each river reach, and done a simulation with 1000 replicates. Methane concentrations have the highest uncertainty and thus are the most sensitive to changes.

In addition, the reviewer comments on the current figure S10, where we compare modelled flux estimates with flux observations. Despite that the comparison is a bit uneven, given that modelled observations have a large model uncertainty and flux observations also have a large measurement uncertainty, the new model output provides a much better validation than before, with an R² of 0.61.

We have also followed the reviewer's advice and partitioned the variance among parameters in the diffusive emissions, by performing an ANOVA on the MC output for each river reach. The new table

S3 summarizes this information and is reproduced below. We did not dare to numerically guess the uncertainty in other processes besides diffusive emissions, so we qualitatively rate the potential uncertainty, as well as provide brief suggestions on how to improve them for future research.

Table S3. Partitioning of uncertainty in the different parameters of diffusive emissions and assessment of other relevant processes and pathways.

Process	Internal variables	% internal uncertainty	Contribution to total uncertainty	Potential future improvements
Diffusive emissions	CH ₄ concentration	49.9 ^a	medium	More reach-scale measurements. Better world coverage across seasons (See figure S1)
	Gas transfer velocity	2.9 ^a	low	Higher spatial resolution hydrological datasets and slope
	River area	47.2 ^a	medium	Better upscaling of river area using remote sensing products
Ebullitive emissions			high	More reach-scale measurements Better understanding of broad-scale drivers.
Methane oxidation			medium	More reach-scale measurements

^a Variance quantified within the Monte Carlo simulation. For each river reach, the results of the Monte Carlo simulation (n=1000) were analyzed using ANOVA, see methods for details. The values shown are global averages.

Aside from this one major comment, my few other comments are relatively minor.

Specific Comments

Figure 1: It's a little surprising to me to see large concentration and emission values in the middle of deserts (e.g. the Sahara and the Arabian desert). I recognize that the hexes have been scaled by runoff to try to mitigate this visual issue. I presume that the actual fluxes from these areas in the calculations are very small (as average discharge is very small), but I think the authors might want to reconsider how they present their data in these figures to better convey the relative balance of fluxes in different parts of the world.

Yes indeed, we assume that the few observations in these biomes have decently high values, despite the low area that is wet. We agree that is something to minimize and hence why we rescaled the hexes by runoff. The reason we use areal fluxes (gCH₄/m² of rivers) is that if we show total emissions (which are then corrected by river area, drying and freezing), areas with large rivers dominate the map, and while it is also an interesting visualization, it would focus the attention towards large rivers. Also, showing an areal rate that anyone can directly compare to their area is maybe easier to comprehend than a much abstract value as total emissions. Given the importance of small streams in general we are afraid that it would be a counter-productive figure. To address this, we tuned a bit more the rescaling of the hexes to minimize even more the deserts.

Line 433: I would rewrite this as “~6900 diffusive flux measurements.” “observations of” seems redundant with “measurements.”

We changed the text as the reviewer suggested

Line 435: Should be “reach,” not “each.”

Thanks for catching this, done

Line 446: Should be “rely” not “relay.”

Done

References used in this review:

1. Biau, G., Scornet, E. A random forest guided tour. TEST 25, 197–227 (2016).
<https://doi.org/10.1007/s11749-016-0481-7>
2. Hopple, A.M., Wilson, R.M., Kolton, M. et al. Massive peatland carbon banks vulnerable to rising temperatures. Nat Commun 11, 2373 (2020).
3. Jansen, J., Thornton, B. F., Wik, M., MacIntyre, S., & Crill, P. M. (2020). Temperature proxies as a solution to biased sampling of lake methane emissions. Geophysical Research Letters, 47, e2020GL088647.
4. Pilla, R. M., Griffiths, N. A., Gu, L., Kao, S.-C., McManamay, R., Ricciuto, D. M., & Shi, X. (2022). Anthropogenically driven climate and landscape change effects on inland water carbon dynamics: What have we learned and where are we going? Global Change Biology, 28, 5601– 5629.
5. Wager, S. T. Hastie, and B. Efron. 2014. Confidence Intervals for Random Forests: The Jackknife and the Infinitesimal Jackknife Journal of Machine Learning Research 15, 1625-1651.
6. Yvon-Durocher, G., Allen, A., Bastviken, D. et al. Methane fluxes show consistent temperature dependence across microbial to ecosystem scales. Nature 507, 488–491 (2014).

Reviewer Reports on the First Revision:

Referees' comments:

Referee #1 (Remarks to the Author):

I would like to express my appreciation to the authors for their effort in revising and improving the manuscript. Several of my comments have been addressed in the revised version, and the authors' response has provided valuable explanation. Nevertheless, I believe that there are still some remaining issues that require further discussion from the authors.

1. In their response, the authors presented their findings from comparing the performance of the random forest model with XGBoost and a one-layer neural network model in terms of R^2 and RMSE. It would be beneficial for the authors to incorporate this discussion into the manuscript (or supplementary materials).

2. I recommend that the authors clarify a few components of their data analyses and reported results:

(i) The authors mention that they tuned the random forest model with data from all the months but then fit separate models for each month using data from related three months. I am curious why the step of tuning with all data was preferred if the authors assume heterogeneity across months or seasons.

(ii) The authors suggest that they chose to use three-month data to have data of a reasonable size. It would be helpful if the authors could clarify how large the resulting dataset is, or how small the monthly dataset is. This would provide reader with a more concrete idea and may be relevant to the performance of the methods random forest, XGBoost, and the neural network model.

(iii) The authors mention that they removed some covariates that were highly correlated. It would be helpful to clarify which covariates were highly correlated and which ones were removed based on what criterion. For example, if covariates A, B, and C are highly correlated, which one or ones of them did the authors choose to retain or remove, and why?

3. It should be noted that feature importance can be obtained not only for random-forest methods but also for other statistical and machine learning methods, including XGBoost and deep neural network (DNN). Therefore, it is not a unique property or advantage of random forests. Similarly, the method for producing uncertainties and confidence or prediction intervals applies to other methods, including XGBoost and some DNN. I would recommend that the authors clarify this point. Otherwise, some readers may have thought these are advantages of random forests.

4. I thank the authors for having included a map of uncertainties in the revised manuscript (Figure S5). However, I have a few issues with this figure: (i) I recommend the authors include the map of predictive standard errors for the concentration. It is more directly related to the confidence and prediction intervals compared to the coefficient of variation (CV). (ii) Figure S5 appears to differ from what we might typically expect to see. We usually would expect to observe higher uncertainty in regions where there is less or no data. In comparison with Figure S1, the uncertainty map in Figure S5 does not appear to indicate this. An explanation is needed.

5. The authors provide additional information on their data aggregation to handle data sets with very different spatial resolutions. However, I believe it is important for the authors to acknowledge and briefly discuss the ecological fallacy, which suggests that the relationship at the aggregated level may be different from the association at a finer or individual level.

6. In Lines 117-120, the authors state that "These sets of fine-scale controls likely drive the roughly ~50% of unexplained variability in our model (Figure S2 and S5), indicating an unresolved discrepancy between global, monthly models and reach-scale or continuous field studies." It is

unclear to me how the authors arrived at the number of approximately 50%. Clarification is needed.

Referee #3 (Remarks to the Author):

Thank you for your careful attention to my comments from the first round of review. I believe the additions of Figure S12 and Table S3 improve our understanding of the sensitivity of of global riverine methane emissions to the various underlying components of the calculation. I have only one addition question in this regard: The caption for Figure S13 suggests that differences in gas transfer velocity could explain the much larger spread in observed diffusive CH₄ fluxes compared to modeled fluxes. However, Figure S12 and Table S3 suggest that results are highly insensitive to differences in gas transfer velocity. These two assertions seem to be incompatible, and I'd like to ask the authors to clarify.

Beyond this concern, I only have a small number of minor suggested edits:

Figure 1: could you include in the legend some information about what runoff values the different grid hexes represent? The caption provides a qualitative assessment, but I think it would be useful to provide a quantitative one as well.

Line 28: hyphen between spatially and explicit can be removed.

Line 33: hyphen should be added between human and dominated.

Line 127: Could you include percentage values in this sentence for >50 and 30-50 deg latitude? It would then be easier to compare to the 37% value from the previous sentence.

Line 672: I would write "which" instead of "and" at the beginning of the last line of the caption.

Author Rebuttals to First Revision:

Referees' comments:

Referee #1 (Remarks to the Author):

I would like to express my appreciation to the authors for their effort in revising and improving the manuscript. Several of my comments have been addressed in the revised version, and the authors' response has provided valuable explanation. Nevertheless, I believe that there are still some remaining issues that require further discussion from the authors.

1. In their response, the authors presented their findings from comparing the performance of the random forest model with XGBoost and a one-layer neural network model in terms of R^2 and RMSE. It would be beneficial for the authors to incorporate this discussion into the manuscript (or supplementary materials).

We have now incorporated the mentioned figure in the Supplementary Materials, together with the methods to develop it, and a brief discussion related to comparing these models. We also refer the reader to this in the methods (line 420-422): “*We used random-forest models to predict CH_4 concentrations and understand the main drivers (but other machine learning models such as XGBoost and a neural network were also explored; see supplementary materials).*”

2. I recommend that the authors clarify a few components of their data analyses and reported results:

(i) The authors mention that they tuned the random forest model with data from all the months but then fit separate models for each month using data from related three months. I am curious why the step of tuning with all data was preferred if the authors assume heterogeneity across months or seasons.

Initially, we tried to fine-tune a model for each month, but the model performance was not very different from the model using the yearly data. Therefore, for the sake of simplicity, we chose to use the same parameters for all the months.

(ii) The authors suggest that they chose to use three-month data to have data of a reasonable size. It would be helpful if the authors could clarify how large the resulting dataset is, or how small the monthly dataset is. This would provide reader with a more concrete idea and may be relevant to the performance of the methods random forest, XGBoost, and the neural network model.

We have now added the information of number of observations for each month in the figure Extended data 1.

(iii) The authors mention that they removed some covariates that were highly correlated. It would be helpful to clarify which covariates were highly correlated and which ones were removed based on what criterion. For example, if covariates A, B, and C are highly correlated, which one or ones of them did the authors choose to retain or remove, and why?

We thank the reviewer to pointing this out. We did not provide the full reasoning as to why each variable was removed to manage the length of the Methods section. However, in this

revision, we now detail the variables that were excluded and provide a general sentence describing the main criteria used (lines 427-433): “...To do this we first removed several independent variables that were highly correlated with others (Pearson $r > 0.95$). We retained the variable that best captured whole ecosystem status or processes. Variables removed were: gross primary production because it was closely related to net primary production; heterotrophic and autotrophic respiration because these were strongly related to total soil respiration; nitrogen load from agriculture, aquaculture and point sources, because these were closely related to phosphorus inputs. Soil properties used also contained a value for the top and bottom soil layer, which were highly related, and so we retained only the top soil properties for modelling.”

3. It should be noted that feature importance can be obtained not only for random-forest methods but also for other statistical and machine learning methods, including XGBoost and deep neural network (DNN). Therefore, it is not a unique property or advantage of random forests. Similarly, the method for producing uncertainties and confidence or prediction intervals applies to other methods, including XGBoost and some DNN. I would recommend that the authors clarify this point. Otherwise, some readers may have thought these are advantages of random forests.

We have now added these points in the supplementary materials where we detail the comparison with other machine learning algorithms: “In addition to the random forest models implemented, we explored whether other machine learning algorithms more oriented towards predictability could improve the results with this dataset. Those methods could in principle provide a more robust prediction and can also be interpreted similarly to random forest models”.

4. I thank the authors for having included a map of uncertainties in the revised manuscript (Figure S5). However, I have a few issues with this figure: (i) I recommend the authors include the map of predictive standard errors for the concentration. It is more directly related to the confidence and prediction intervals compared to the coefficient of variation (CV). (ii) Figure S5 appears to differ from what we might typically expect to see. We usually would expect to observe higher uncertainty in regions where there is less or no data. In comparison with Figure S1, the uncertainty map in Figure S5 does not appear to indicate this. An explanation is needed.

We have made this change as the reviewer suggested, showing now the SE of the prediction. The pattern now is as the reviewer describes, with more uncertain observations where the model applicability is poor (compare with Extended data 4).

5. The authors provide additional information on their data aggregation to handle data sets with very different spatial resolutions. However, I believe it is important for the authors to acknowledge and briefly discuss the ecological fallacy, which suggests that the relationship at the aggregated level may be different from the association at a finer or individual level.

We have now added a more explicit statement and an acknowledgement of the ecological fallacy in the methods: “We acknowledge that this data aggregation procedure can result in relationships at large spatial scales that may be different when assessing the same relationships at finer spatial scales, the so-called ecological fallacy, but this was a necessary step to quantify and understand river emissions at global scales.” (lines 399-402)

6. In Lines 117-120, the authors state that “These sets of fine-scale controls likely drive the roughly ~50% of unexplained variability in our model (Figure S2 and S5), indicating an unresolved discrepancy between global, monthly models and reach-scale or continuous field studies.” It is unclear to me how the authors arrived at the number of approximately 50%. Clarification is needed.

With the 50% we were referring to the R^2 of the prediction vs modelled data, which is around 0.5. However, we are aware this assumption is not entirely correct. Thus, we have now changed the text so that it does not contain a quantitative assessment, but a more general statement: “These sets of fine-scale controls likely drive the substantial fraction of unexplained variability in our model” (line 112).

Referee #3 (Remarks to the Author):

Thank you for your careful attention to my comments from the first round of review. I believe the additions of Figure S12 and Table S3 improve our understanding of the sensitivity of global riverine methane emissions to the various underlying components of the calculation. I have only one addition question in this regard: The caption for Figure S13 suggests that differences in gas transfer velocity could explain the much larger spread in observed diffusive CH₄ fluxes compared to modeled fluxes. However, Figure S12 and Table S3 suggest that results are highly insensitive to differences in gas transfer velocity. These two assertions seem to be incompatible, and I'd like to ask the authors to clarify.

We thank the reviewer for this comment. The reason for the discrepancy is that the uncertainty in field empirical measurements of fluxes and the gas transfer velocity is very high (Hall and Ulseth, 2019). The uncertainty we use in our model originates from the model relating the gas transfer velocity with hydraulic properties from Raymond et al. (2012), which reports the SD of the confidence interval. That error measure is relatively small compared to a prediction interval and thus we are likely overestimating the uncertainty of the modelled gas transfer velocity. Put another way, as in the sensitivity analysis, we have used a Monte Carlo simulation by subtracting or adding 1 SD to the values, the results from this will heavily depend on the uncertainty of the original parameters, which is much higher for CH₄ than for k. We have now added some text discussing this in the caption of Extended data 8.

Beyond this concern, I only have a small number of minor suggested edits:

Figure 1: could you include in the legend some information about what runoff values the different grid hexes represent? The caption provides a qualitative assessment, but I think it would be useful to provide a quantitative one as well.

Given the small size of the hexagons (hexes) in the map it would be hard to appreciate the legend properly. Instead, we provide one sentence with the thresholds used in the caption: “. Areas with runoff $>1500 \text{ mm yr}^{-1}$ have a full-sized hexagon, hexagons in areas with runoff of 500 mm yr^{-1} have been reduced by 10%, and hexes with a runoff below 50 mm yr^{-1} have been reduced by 50%.”

Line 28: hyphen between spatially and explicit can be removed.

Changed as suggested by the reviewer

Line 33: hyphen should be added between human and dominated.

Changed as suggested by the reviewer

Line 127: Could you include percentage values in this sentence for >50 and 30-50 deg latitude? It would then be easier to compare to the 37% value from the previous sentence.

We have now added the value in lines 122-123.

Line 672: I would write "which" instead of "and" at the beginning of the last line of the caption.

Thanks, we have changed it as suggested.

References

Hall Jr, R. O., & Ulseth, A. J. (2020). Gas exchange in streams and rivers. Wiley Interdisciplinary Reviews: Water, 7(1), e1391.

Raymond, P. A., Zappa, C. J., Butman, D., Bott, T. L., Potter, J., Mulholland, P., ... & Newbold, D. (2012). Scaling the gas transfer velocity and hydraulic geometry in streams and small rivers. Limnology and Oceanography: Fluids and Environments, 2(1), 41-53.